# Disparate Impact in Differential Privacy from Gradient Misalignment

**Maria S. Esipova, Atiyeh Ashari Ghomi, Yaqiao Luo & Jesse C. Cresswell**
Layer 6 AI
{maria, atiyeh, emily, jesse}@layer6.ai

## Abstract

As machine learning becomes more widespread throughout society, aspects including data privacy and fairness must be carefully considered, and are crucial for deployment in highly regulated industries. Unfortunately, the application of privacy enhancing technologies can worsen unfair tendencies in models. In particular, one of the most widely used techniques for private model training, differentially private stochastic gradient descent (DPSGD), frequently intensifies disparate impact on groups within data. In this work we study the fine-grained causes of unfairness in DPSGD and identify gradient misalignment due to inequitable gradient clipping as the most significant source. This observation leads us to a new method for reducing unfairness by preventing gradient misalignment in DPSGD.

## 1 Introduction

The increasingly widespread use of machine learning throughout society has brought into focus social, ethical, and legal considerations surrounding its use. In highly regulated industries, such as healthcare and banking, regional laws and regulations require data collection and analysis to respect the privacy of individuals.[1] Other regulations focus on the fairness of how models are developed and used.[2] As machine learning is progressively adopted in highly regulated industries, the privacy and fairness aspects of models must be considered at all stages of the modelling lifecycle.

There are many privacy enhancing technologies including differential privacy (Dwork et al., 2006), federated learning (McMahan et al., 2017), secure multiparty computation (Yao, 1986), and homomorphic encryption (Gentry, 2009) that are used separately or jointly to protect the privacy of individuals whose data is used for machine learning (Choquette-Choo et al., 2020; Adnan et al., 2022; Kalra et al., 2021). The latter three technologies find usage in sharing schemes and can allow data to be analysed while preventing its exposure to the wrong parties. However, the procedures usually return a trained model which itself can leak private information (Carlini et al., 2019). On the other hand, differential privacy (DP) focuses on quantifying the privacy cost of disclosing aggregated information about a dataset, and can guarantee that nothing is learned about individuals that could not be inferred from population-level correlations (Jagielski et al., 2019). Hence, DP is often used when the results of data analysis will be made publicly available, for instance when exposing the outputs of a model, or the results of the most recent US census (Abowd, 2018).

Not only must privacy be protected for applications in regulated industries, models must be fair. While there is no single definition that captures what it means to be fair, with regards to model-based decision making fairness may preclude disparate treatment or disparate impact (Mehrabi et al., 2021). Disparate treatment is usually concerned with how models are applied across populations, whereas disparate impact can arise from biases in datasets that are amplified by the greedy nature of loss minimization algorithms (Buolamwini & Gebru, 2018). Differences in model performance across protected groups can result in a significant negative monetary, health, or societal impact for individuals who are discriminated against (Chouldechova & Roth, 2020).

---

[1]Examples of laws governing data privacy include the General Data Protection Regulation in Europe, Health Insurance Portability and Accountability Act in the USA, and Personal Information Protection and Electronic Documents Act in Canada.

[2]In the USA, fair lending laws including the Fair Housing Act, and Equal Credit Opportunity Act prohibit discrimination based on protected characteristics such as race, age, and sex.

Unfortunately, it has been observed that disparate impact can be exacerbated by applying DP in machine learning (Bagdasaryan et al., 2019). Applications of DP always come with a privacy-utility tradeoff, where stronger guarantees of privacy negatively impact the usefulness of results - model performance in this context (Dwork & Roth, 2014). Underrepresented groups within the population can experience disparity in the cost of adding privacy, hence, fairness concerns are a major obstacle to deploying models trained with DP.

The causes of unfairness in DP depend on the techniques used, but are not fully understood. For the most widely used technique, differentially private stochastic gradient descent (DPSGD), two sources of error are introduced that impact model utility. Per-sample gradients are clipped to a fixed upper bound on their norm, then noise is added to the averaged gradient. Disparate impact from DPSGD was initially hypothesized to be rooted in unbalanced datasets (Bagdasaryan et al., 2019), though counterexamples were found by Xu et al. (2021). Recent research claims disparate impact to be caused by incommensurate clipping errors across groups, in turn effected by a large difference in average group gradient norms (Xu et al., 2021; Tran et al., 2021a).

In this work we highlight the disparate impact of gradient misalignment. In particular, we claim that the most significant cause of disparate impact is the difference in the direction of the unclipped and clipped gradients, which in turn can be caused by aggressive clipping and imbalances of gradient norms between groups. Our analysis of direction errors leads to a variant of DPSGD with properly aligned gradients. We explore this alternate method in relation to disparate impact and show that it not only significantly reduces the cost of privacy across all protected groups, it also reduces the *difference* in cost of privacy for all groups. Hence, it removes disparate impact and is more effective than previous proposals in doing so. On top of this, it is the only approach which does not require access to protected group labels, and thereby avoids disparate treatment of groups. In summary we:

- Conduct a more fine-grained analysis of disparate impact in DPSGD, and demonstrate gradient misalignment to be the most significant cause;
- Identify an existing algorithm, previously undiscussed in the fairness context, which properly aligns gradients, and show it reduces disparate impact and disparate treatment;
- Improve the utility of said algorithm via two alterations;
- Experimentally verify that aligning gradients is more successful at mitigating disparate impact than previous approaches.

## 2 RELATED WORK

**Privacy and Fairness:** While privacy and fairness have been extensively studied separately, recently their interactions have come into focus. Ekstrand et al. (2018) considered the intersection of privacy and fairness for several definitions of privacy. This research gained new urgency when Bagdasaryan et al. (2019) observed that DPSGD exacerbated existing disparity in model accuracy on underrepresented groups. Disparate impact due to DP was further observed in Pujol et al. (2020) and Farrand et al. (2020) for varying levels of group imbalance. Using an adversarial definition of privacy, Jaiswal & Mower Provost (2020) found that overrepresented groups can incur higher privacy costs. Similar examples were shown in Xu et al. (2021) for DPSGD, and disparate impact was linked to groups having larger gradient norms.

Other fairness-aware learning research has evaluated the fairness of a private model's outcomes on protected groups. In this context fairness might refer to a statistical condition of non-discrimination with respect to groups (Mozannar et al., 2020; Tran et al., 2021b), for example, equalized odds (Jagielski et al., 2019), equality of opportunity (Cummings et al., 2019), or demographic parity (Xu et al., 2019; Farrand et al., 2020). Chang & Shokri (2021) empirically found that imposing fairness constraints on private models could lead to higher privacy loss for certain groups. We consider cross-model fairness where the *cost of adding privacy* to a non-private model must be fairly distributed between groups.

**Adaptive Clipping:** Many variations on the clipping procedure in DPSGD have been proposed to improve properties other than fairness. Adaptive clipping comes in many forms, but usually tunes the clipping threshold during training to provide better privacy-utility tradeoffs and convergence (Andrew et al., 2021; Pichapati et al., 2019). The convergence of DPSGD connects to the symmetry properties of the distribution of gradients (Chen et al., 2020) which are affected by clipping.

## 3 BACKGROUND

### 3.1 SETTING AND DEFINITIONS

We begin by laying out the problem setting and review the relevant definitions for discussing fairness in privacy. For concreteness we consider a binary classification problem on a dataset $D$ which consists of $n$ points of the form $(x_i, a_i, y_i)$, where $x_i \in \mathbb{R}^d$ is a feature vector, $y_i \in \{0, 1\}$ is a binary label, and $a_i \in [K]$ refers to a protected group attribute which partitions the data. The group label $a_i$ can optionally be an attribute in $x_i$, the label value $y_i$, or some distinct auxiliary value.

The goal is to train a model $f_\theta : \mathbb{R}^d \rightarrow [0, 1]$ with parameter vector $\theta$ that is simultaneously useful and private, and in which the application of privacy is fair. Utility in the empirical risk minimization (ERM) problem is governed by the per-sample loss $\ell : [0, 1] \times \{0, 1\} \rightarrow \mathbb{R}$, with the optimal model minimizing the objective $\mathcal{L}(\theta; D) = \frac{1}{n} \sum_{i \in D} \ell(f_\theta(x_i), y_i)$, which happens for optimal parameters $\theta^* = \arg\min_\theta \mathcal{L}(\theta; D)$. The requirement of privacy is applied to the model through its parameters; private parameters $\tilde{\theta}$ must be obtained while exposing a minimal amount of private information in $D$. For this we apply the framework of differential privacy, recounted in the next section.

Fairness of the privacy methodology can be measured in terms of the disparate impact that applying privacy has on the protected groups. As in Bagdasaryan et al. (2019), we use a version of *accuracy parity*, the difference in classification accuracy across protected groups after adding privacy. We denote a subset of the data containing all points belonging to group $k$ as $D_k = \{(x_i, a_i, y_i) \in D \mid a_i = k\}$. A private model has accuracy parity for subset $D_k$ if it minimizes the *privacy cost*

$$\pi(\theta, D_k) = \mathrm{acc}(\theta^*; D_k) - \mathbb{E}_{\tilde{\theta}}[\mathrm{acc}(\tilde{\theta}; D_k)], \tag{1}$$

where the expectation is over the randomness involved in acquiring private model parameters. Of course, metrics other than classification accuracy could be used as required by the problem setting. Alternatively, fairness for privacy can be measured at the level of the loss function as in Tran et al. (2021a), which is more amenable to analyzing the causes of unfairness. The *excessive risk* over the course of training experienced by a group is

$$R(\theta, D_k) = \mathbb{E}_{\tilde{\theta}}[\mathcal{L}(\tilde{\theta}; D_k)] - \mathcal{L}(\theta^*; D_k). \tag{2}$$

When the model is clear from context we denote $R(\theta; D_k)$ as $R_k$, and similarly for privacy cost $\pi_k$. For both accuracy and loss we consider the gap between disparate impact values across groups. The *privacy cost gap* is $\pi_{a,b} = |\pi_a - \pi_b|$ for groups $a, b \in [K]$, and the *excessive risk gap* refers to $R_{a,b} = |R_a - R_b|$. The goal of a fair private classifier is to minimize the privacy cost and/or excessive risk for all values of the protected group attribute, while maintaining small fairness gaps.

### 3.2 DIFFERENTIAL PRIVACY

Differential privacy (DP) (Dwork et al., 2006) is a widely used framework for quantifying the privacy consumed by a data analysis procedure. Formally, let $D$ represent a set of data points, and $M$ a probabilistic function, or *mechanism*, acting on datasets. We say that the mechanism is $(\epsilon, \delta)$-*differentially private* if for all subsets of possible outputs $S \subseteq \mathrm{Range}(M)$, and for all pairs of databases $D$ and $D'$ that differ by the addition or removal of one element,

$$\Pr[M(D) \in S] \leq \exp(\epsilon) \Pr[M(D') \in S] + \delta. \tag{3}$$

For the ERM problem, there are several ways to train a differentially private model (Chaudhuri et al., 2011). In this work we consider models that can be

---

**Algorithm 1** DPSGD

**Require:** Iterations $T$, Dataset $D$, sampling rate $q$, clipping bound $C_0$, noise multiplier $\sigma$, learning rates $\eta_t$

Initialize $\theta_0$ randomly
**for** $t$ in $0, \ldots, T - 1$ **do**
    $B \leftarrow$ Poisson sample of $D$ with rate $q$
    **for** $(x_i, y_i)$ in $B$ **do**
        $g_i \leftarrow \nabla_\theta \ell(f_{\theta_t}(x_i), y_i)$
        $\bar{g}_i \leftarrow g_i \cdot \min\left(1, \frac{C_0}{\|g_i\|}\right)$
    $\tilde{g}_B \leftarrow \frac{1}{|B|}\left(\sum_{i \in B} \bar{g}_i + \mathcal{N}(0, \sigma^2 C_0^2 \mathbb{I})\right)$
    $\theta_{t+1} \leftarrow \theta_t - \eta_t \tilde{g}_B$

---

trained with stochastic gradient descent (SGD), such as neural networks, and focus on the most successful approach, DPSGD (Abadi et al., 2016), in which the Gaussian mechanism (Dwork & Roth, 2014) is applied to gradient updates as in Alg. 1. Since per-sample gradients $g_i$ generally do not have finite sensitivity, defined as $\Delta_h = \max_{D,D'} \|h(D) - h(D')\|$ for a function $h$, they are first clipped to have norm upper bounded by a fixed hyperparameter $C_0$. Clipped gradients $\bar{g}_i$ in a batch $B \subset D$ are aggregated into $\bar{g}_B$ and noise is added to produce $\tilde{g}_B$ used in the parameter update.

### 3.3 Fairness concerns from clipping and noise in DPSGD

The two most significant steps in DPSGD, clipping and adding noise, can impact the learning process disproportionately across groups, but the exact conditions where disparate impact will occur have been debated (Bagdasaryan et al., 2019; Farrand et al., 2020; Xu et al., 2021; Tran et al., 2021a). The most concrete connection so far appears in (Tran et al., 2021a), where the expected loss $\mathcal{L}(\theta; D_a)$ is decomposed into terms contributing to the excessive risk at a single iteration for group $a$, $R_a$:

**Proposition 1** (Tran et al. (2021a)). *Consider the ERM problem with twice-differentiable loss $\ell$ with respect to the model parameters. The expected loss $\mathbb{E}[\mathcal{L}(\theta_{t+1}; D_a)]$ of group $a \in [K]$ at iteration $t$ is approximated up to second order in $\|\theta_{t+1} - \theta_t\|$ as:*

$$\mathbb{E}[\mathcal{L}(\theta_{t+1}; D_a)] \approx \mathcal{L}(\theta_t; D_a) - \eta_t\langle g_{D_a}, g_D\rangle + \frac{\eta_t^2}{2}\mathbb{E}[g_B^T H_\ell^a g_B] \qquad \text{(non-private term)}$$

$$+ \eta_t\langle g_{D_a}, g_D - \bar{g}_D\rangle + \frac{\eta_t^2}{2}\left(\mathbb{E}[\bar{g}_B^T H_\ell^a \bar{g}_B] - \mathbb{E}[g_B^T H_\ell^a g_B]\right) \qquad (R_a^{\text{clip}})$$

$$+ \frac{\eta_t^2}{2}\text{Tr}(H_\ell^a)C_0^2\sigma^2. \qquad (R_a^{\text{noise}})$$

*The expectation is taken over the randomness of the DP mechanisms, and batches of data.*

Terms in the first line appear for ordinary SGD, and do not contribute to the excessive risk Eq. (2). The terms in the second line, $R_a^{\text{clip}}$, are caused by clipping since they cancel when $\bar{g}_B = g_B$ for every batch. They involve gradients $g_{D_a}$ and Hessians $H_\ell^a$, averaged over datapoints belonging to group $a$. The final term, $R_a^{\text{noise}}$, depends on the scale of noise added in Alg. 1, as well as the trace of the Hessian, also called the Laplacian, averaged over $D_a$. Based on Prop. 1, Tran et al. (2021a) argue that clipping causes excessive risk to groups with large gradient norms, which can result from large input norms $\|x_i\|$. Whether or not a group is underrepresented has less influence. In the next section we provide a new perspective on $R_a^{\text{clip}}$ and the underlying causes of unfairness in DPSGD.

## 4 Disparate impact is caused by gradient misalignment

Clipping in DPSGD introduces two types of error to the clipped batch gradient $\bar{g}_B$. It will generally have different norm than $\|g_B\|$, and be misaligned compared to the SGD batch gradient, $g_B$. At a high level, gradient misalignment poses a more serious problem to the convergence of DPSGD than magnitude error, as illustrated in Fig. 1. Changing only the norm means gradient descent will still step towards the (local) minimum of the loss function, and any norm error could be completely compensated for by adapting the learning rate $\eta_t$. In contrast, a misaligned gradient could result in a

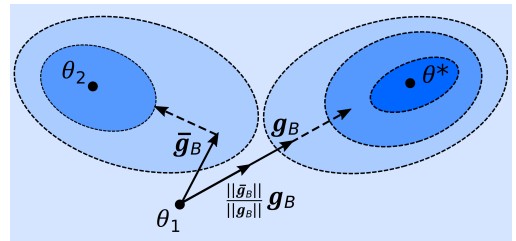

Figure 1: Direction errors from clipping are more severe than magnitude errors over the course of training and can lead to suboptimal convergence.

step towards significantly worse regions of the loss landscape causing catastrophic failures of convergence. Misaligned gradients add bias which compounds over training, as underrepresented or complex groups are systematically clipped. For comparison, adding noise to the aggregated gradient does not add bias, so noise errors tend to cancel out over training. We aim to quantify the relative impact of these effects and how they contribute to the excessive risk.

We can distinguish the effects of clipping by rewriting the clipped batch gradient as $\bar{g}_B = M_B\left(\frac{\|\bar{g}_B\|}{\|g_B\|}g_B\right)$ for an orthogonal matrix $M_B$ such that $\bar{g}_B$ and $M_B g_B$ are colinear. As a proof of concept that gradient misalignment is the more severe error we compared models trained by taking steps $\frac{\|\bar{g}_B\|}{\|g_B\|}g_B$ vs. $M_B g_B$ with no noise added. These represent magnitude errors and direction errors from clipping, respectively. The models were trained on MNIST with class 8 undersampled, and the results compare the typical class 2 to the underrepresented class 8; full details are provided in App. B. As seen in Table 1, direction error is more detrimental to performance than magnitude error. In particular, it disproportionately increases loss and decreases accuracy on the underrepresented class 8.

Table 1: Effect of direction vs. magnitude error on MNIST with class 8 undersampled. The results compare accuracy and loss on the typical class 2 to the underrepresented class 8.

| Type of error | Acc 2 | Acc 8 | Loss 2 | Loss 8 |
|---|---|---|---|---|
| Magnitude | 99.0 | 93.5 | 0.002 | 0.005 |
| Direction | 96.8 | 84.1 | 0.076 | 0.518 |

Our first theoretical result quantifies the excessive risk from the two types of errors, and follows from a Taylor expansion of the expected loss using $\bar{g}_B$ in the gradient descent update compared to $g_B$. The excessive risk from magnitude error comes from comparing $g_B$ to $\frac{\|\bar{g}_B\|}{\|g_B\|}g_B$, while that of gradient misalignment is isolated by comparing $\bar{g}_B = M_B\left(\frac{\|\bar{g}_B\|}{\|g_B\|}g_B\right)$ to $\frac{\|\bar{g}_B\|}{\|g_B\|}g_B$ (see Fig. 1).

**Proposition 2.** *Consider the ERM problem with twice-differentiable loss $\ell$ with respect to the model parameters. The excessive risk due to clipping experienced by group $a \in [K]$ at iteration $t$ is approximated up to second order in $\|\theta_{t+1} - \theta_t\|$ as*

$$R_a^{\mathrm{clip}} \approx \eta_t \left\langle g_{D_a}, \mathbb{E}\left[\left(1 - \frac{\|\bar{g}_B\|}{\|g_B\|}\right)g_B\right]\right\rangle + \frac{\eta_t^2}{2}\mathbb{E}\left[\left(\frac{\|\bar{g}_B\|^2}{\|g_B\|^2} - 1\right)g_B^T H_\ell^a g_B\right] \qquad (R_a^{\mathrm{mag}})$$

$$+ \eta_t \left\langle g_{D_a}, \mathbb{E}\left[\frac{\|\bar{g}_B\|}{\|g_B\|}(g_B - M_B g_B)\right]\right\rangle + \frac{\eta_t^2}{2}\mathbb{E}\left[\frac{\|\bar{g}_B\|^2}{\|g_B\|^2}\left((M_B g_B)^T H_\ell^a (M_B g_B) - g_B^T H_\ell^a g_B\right)\right], \quad (R_a^{\mathrm{dir}})$$

*where $g_{D_a}$, $\bar{g}_{D_a}$ denote the average non-clipped and clipped gradients over group $a$ at iteration $t$, $H_\ell^a$ refers to the Hessian over group $a$, and $M_B$ is an orthogonal matrix such that $\bar{g}_B$ and $M_B g_B$ are colinear. The expectations are taken over batches of data.*

We provide a derivation in App. A. Note that when the magnitude error is zero for all batches, $\|g_B\| = \|\bar{g}_B\|$, we have that $R_a^{\mathrm{mag}} = 0$ as expected. As well, when there is no gradient misalignment then $M_B$ is the identity matrix for every batch, and so $R_a^{\mathrm{dir}} = 0$.

To determine the characteristics of groups that will have unfair outcomes from clipping in DPSGD we can distill a simpler condition for when $R_a^{\mathrm{dir}} > R_b^{\mathrm{dir}}$. Tran et al. (2021a) already provide such a condition for clipping overall, however it does not effectively account for the danger of gradient misalignment. Their condition is sufficient, but not necessary, and some of its looseness stems from the inequality $x^T y \geq -\|x\|\|y\|$ used to convert all terms in $R_a^{\mathrm{clip}}$ into expressions involving group gradient norms. This approach loses information about gradient direction. We instead propose a tighter analysis of $R_a^{\mathrm{dir}} - R_b^{\mathrm{dir}}$ using $x^T y = \|x\|\|y\|\cos\theta$, where $\theta = \angle(x, y)$.

**Proposition 3.** *Assume the loss $\ell$ is twice continuously differentiable and convex with respect to the model parameters. As well, assume that $\eta_t \leq (\max_{k \in [K]} \lambda_k)^{-1}$ where $\lambda_k$ is the maximum eigenvalue of the Hessian $H_\ell^k$. For groups $a, b \in [K]$, $R_a^{\mathrm{dir}} > R_b^{\mathrm{dir}}$ if*

$$\mathbb{E}\left[\|\bar{g}_B\|(\cos\theta_B^a - \cos\bar{\theta}_B^a)\right] > \frac{\|g_{D_b}\|}{\|g_{D_a}\|}\mathbb{E}\left[\|\bar{g}_B\|(\cos\theta_B^b - \cos\bar{\theta}_B^b)\right] + \frac{\mathbb{E}[\|\bar{g}_B\|^2]}{\|g_{D_a}\|}, \qquad (4)$$

*where $\theta_B^k = \angle(g_{D_k}, g_B)$ and $\bar{\theta}_B^k = \angle(g_{D_k}, \bar{g}_B)$ for a group $k \in [K]$. Furthermore, the bound is tight.*

App. A contains our proof. Prop. 3 shows that if the clipping operation disproportionately and sufficiently increases the direction error for group $a$ relative to group $b$, then group $a$ incurs larger excessive risk due to gradient misalignment.

The lower bound for $R_a^{\mathrm{dir}} - R_b^{\mathrm{dir}}$ inferred from Eq. 4 is tight, and in our experiments we empirically show that it is close to saturation in a typical case. Hence, when the direction errors for groups $a$ and $b$ are small (i.e. we expect that $\theta_B^i \approx \bar{\theta}_B^i$ for $i = a, b$), we have that $R_a^{\mathrm{dir}} - R_b^{\mathrm{dir}} \approx 0$ regardless of the size of $\|g_{D_a}\|$ relative to $\|g_{D_b}\|$. It follows that *clipping does not negatively impact excessive risk if gradients remain aligned.* On the other hand if direction error is not close to zero, large group gradient norms do exacerbate the error in direction, as the dominant term of $R_a^{\mathrm{dir}}$ scales with $\|g_{D_a}\|$.

The excessive risk in Eq. 2 is evaluated at the end of training, whereas Props. 1 and 2 estimate it per-iteration. Fig. 1 demonstrates that the full impact of clipping errors may not be felt per-iteration, but only at convergence. Indeed what matters to the end user is how fair the final model is, not how fair any intermediate training step is. However, it is not possible to attribute overall excessive risk to the per-iteration terms $R_a^{\mathrm{dir}}$, $R_a^{\mathrm{mag}}$, and $R_a^{\mathrm{noise}}$, since the optimal $\theta_i^*$ used in the expansions of Props. 1 and 2 differ at each iteration, and do not equal the overall optimal $\theta^*$. Still, Table 1 demonstrates that gradient misalignment is the main cause of disparate impact, so we seek a method to prevent it.

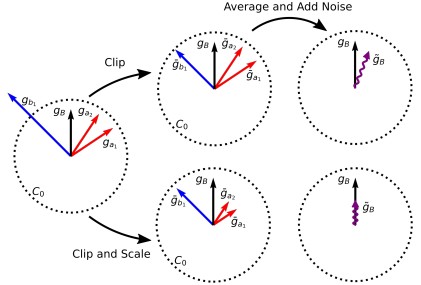
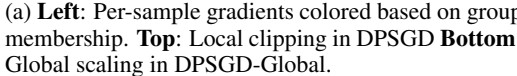

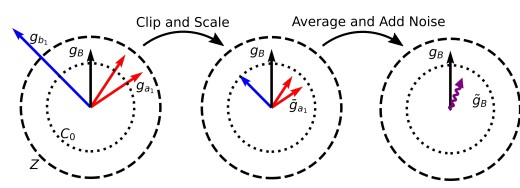

(a) **Left**: Per-sample gradients colored based on group membership. **Top**: Local clipping in DPSGD **Bottom**: Global scaling in DPSGD-Global.

(b) In DPSGD-Global-Adapt scaling alone does not guarantee finite sensitivity, so gradients with norm above $Z$ are clipped to $C_0$ (DPSGD-Global clips large gradients to 0 rather than $C_0$).

Figure 2: Illustration of privatization steps in DPSGD, DPSGD-Global, and DPSGD-Global-Adapt

## 5 PREVENTING GRADIENT MISALIGNMENT IN DPSGD

Our results so far show that gradient misalignment due to clipping is the most significant cause of unfairness in DPSGD. Logically, $R_a^{\text{dir}}$ would be minimized if privatization left the direction of $g_B$ unchanged. A promising avenue is to *scale down all per-sample gradients in a batch by the same amount*. This is the approach taken by DPSGD-Global (Bu et al., 2021), which was recently proposed to improve the convergence of DPSGD, and has not been discussed in the context of fairness before. Our theoretical results suggest that global scaling will reduce disparate impact.

DPSGD-Global (Alg. 2) aims to preserve privacy by scaling gradients as $\bar{g}_i = \gamma g_i$, $0 < \gamma < 1$. Of course, scaling alone is insufficient to ensure per-sample gradients have bounded sensitivity. However, supposing that there were a strict upper bound $Z \geq \|g_i\| \, \forall i \in D$, then scaling all gradients by $\gamma = C_0/Z$ would guarantee bounded sensitivity of $C_0$ for each $\bar{g}_i$ (Fig. 2a). Given sufficient smoothness of the loss function, for any sample of data there will be such an upper bound $\max_{i \in D} \|g_i\|$, but determining it exactly cannot be done in a differentially private manner. DPSGD-Global sets $Z$ as a hyper-

---

**Algorithm 2** DPSGD-Global(-Adapt)

**Require:** Iterations $T$, Dataset $D$, sampling rate $q$, clipping bound $C_0$, strict clipping bound $Z \geq C_0$, noise multipliers $\sigma_1$, ($\sigma_2$), learning rates $\eta_t$, (clipping learning rate $\eta_Z$, threshold $\tau \geq 0$)

Initialize $\theta_0$ randomly
**for** $t$ in $0, \ldots, T-1$ **do**
$\quad B \leftarrow$ Poisson sample of $D$ with rate $q$
$\quad$**for** $(x_i, y_i)$ in $B$ **do**
$\quad\quad g_i \leftarrow \nabla_\theta \ell(f_{\theta_t}(x_i), y_i)$
$\quad\quad \gamma_i \leftarrow \begin{cases} \frac{C_0}{Z}, & \|g_i\| \leq Z \\ 0 \; (\frac{C_0}{\|g_i\|}), & \|g_i\| > Z \end{cases}$
$\quad\quad \bar{g}_i \leftarrow \gamma_i g_i$
$\quad \tilde{g}_B \leftarrow \frac{1}{|B|} \left( \sum_{i \in B} \bar{g}_i + \mathcal{N}(0, \sigma_1^2 C_0^2 \mathbb{I}) \right)$
$\quad \theta_{t+1} \leftarrow \theta_t - \eta_t \tilde{g}_B$
$\quad$(Adaptively set $Z$):
$\quad b_t \leftarrow |\{i : \|g_i\| > \tau \cdot Z\}|$
$\quad \tilde{b}_t \leftarrow \frac{1}{|B|}(b_t + \mathcal{N}(0, \sigma_2^2))$
$\quad Z \leftarrow Z \cdot \exp(-\eta_Z + \tilde{b}_t)$

---

parameter without looking at the data, in the same way $C_0$ is chosen in DPSGD. If $Z$ fails to be a strict upper bound, any gradients with $\|g_i\| > Z$ are discarded to guarantee a bound on sensitivity. When $Z$ is chosen sufficiently large, no gradients are discarded and gradient misalignment is avoided. The drawback of a large $Z$ is that the scaled gradients $\bar{g}_i$ will become small and convergence of gradient descent may be hindered.

In addition to identifying that DPSGD-Global has the potential to reduce disparate impact, we propose two modifications to improve its utility. First, we note that discarding gradients with $\|g_i\| > Z$ can exacerbate disparate impact as it is often underrepresented groups that have large gradient norms (Xu et al., 2021). Instead, we clip large gradients to have norm $C_0$, which preserves more information while maintaining finite sensitivity (Fig. 2b). Second, rather than choosing $Z$ as a hyperparameter, we adaptively update $Z$ to upper-bound $\max_{i \in B} \|g_i\|$. When $Z$ is larger than all gradients it should be reduced to scale down gradients less, but if gradients are being clipped, $Z$ should be increased. $Z$ can be updated each iteration by privately estimating $b_t$, the number of gradients in $B$ that are larger than $Z$ times a tolerance threshold $\tau \geq 0$. Since $b_t$ is a unit sensitivity quantity we can estimate it privately as $\tilde{b}_t = \frac{1}{|B|}(b_t + \mathcal{N}(0, \sigma_2^2))$. Then, we use the geometric update rule $Z \leftarrow Z \cdot \exp(-\eta_Z + \tilde{b}_t)$ with a learning rate $\eta_Z$ (cf. (Andrew et al., 2021)). When all samples

have gradient norm less than or equal to $\tau \cdot Z$, then in expectation $\tilde{b}_t = 0$ and $Z$ is decreased by a factor of $\exp(-\eta_Z)$. Alternatively, $Z$ is increased when $\tilde{b}_t > \eta_Z$, which occurs with probability 0.977 when $\frac{b_t}{|B|} \geq \eta_Z + \frac{2\sigma_2}{|B|}$. As a result, with high probability the algorithm will not have more than $|B|\eta_Z + 2\sigma_2$ gradients with norm exceeding $\tau \cdot Z$.

We call the method with our two alterations DPSGD-Global-Adapt, shown in Alg. 2 in red parentheses. We empirically find in Sec. 6 that both global approaches improve fairness compared to prior methods, and that DPSGD-Global-Adapt has improved utility over DPSGD-Global. While the alterations are minor, our main contributions are elucidating that gradient misalignment is the main cause of disparate impact, and identifying that global scaling can prevent this problem.

Both global methods apply the sampled Gaussian mechanism (Mironov et al., 2019) to gradient norms with a sensitivity of $C_0$, and hence are amenable to the same DP analysis as DPSGD itself. In DPSGD-Global-Adapt, the additional step of privately estimating the number of gradients with norm larger than $\tau \cdot Z$ must be accounted for in the overall DP guarantee via a composition of sampled Gaussian mechanisms. From the analysis in (Mironov et al., 2019), DPSGD-Global-Adapt is $(\epsilon, \delta)$-DP for any $\sigma_1, \sigma_2 > 0$, where $\epsilon$ can be determined numerically given $\delta$. However, our adaptive method is empirically not sensitive to the exact count $b_t$, so a relatively large amount of noise can be used, see (Andrew et al., 2021) for comparison. In practice we used $\sigma_2 \approx 10\sigma_1$ which produced a negligible additional cost in the overall privacy budget.

Finally, we note that other approaches for mitigating unfairness, specifically DPSGD-F (Xu et al., 2021) and that of Tran et al. (2021a), require protected group labels for the training set. Collecting such labels may expose individuals to additional privacy risks in the case of security breaches, or may be prohibited in practice. Both global methods have the advantage of not requiring protected group labels for training data, and treat all training examples on an equal footing, thereby avoiding disparate treatment, while disparate impact is mitigated by reducing gradient misalignment.

## 6 EXPERIMENTS

In our experiments we provide evidence that gradient misalignment is the most significant cause of unfairness, and demonstrate that global scaling can effectively reduce unfairness by aligning gradients. Our code for reproducing the experiments is provided as supplementary material.

### 6.1 EXPERIMENT SETTINGS

For all experiments, full details are provided in App. B. We use an artificially unbalanced MNIST training dataset where class 8 only constitutes about 1% of the dataset on average, and protected groups are the classes. We also use two census datasets popular in the ML fairness literature, Adult and Dutch (van der Laan, 2000), preprocessed as in Le Quy et al. (2022). For both datasets, "sex" is the protected group attribute which is balanced between males and females. Finally, we use the CelebA dataset (Liu et al., 2015) for binary classification on the gender label. The protected group attribute is whether the image contains eyeglasses. Images with eyeglasses comprise 12% of male images but only 2% of female images, and are more difficult to classify accurately.

We compare both global scaling techniques (Alg. 2) against two methods designed to reduce unfairness, DPSGD-F (Xu et al., 2021) (Alg. 3) and the Fairness-Lens method (Tran et al., 2021a) (Alg. 4), both of which are reviewed in App. B.5. Each method's effectiveness in removing disparate impact is measured using privacy cost $\pi_a$ (Eq. 1), and excessive risk $R_a$ (Eq. 2) per group, as well as the privacy cost gap $\pi_{a,b}$, and excessive risk gap $R_{a,b}$ between groups. For MNIST, the underrepresented group 8 is compared to group 2 (Xu et al., 2021). All experiments were run for 5 random seeds, and results are given as means $\pm$ standard errors.

For MNIST and CelebA, all methods train a convolutional neural network with two layers of 32 and 16 channels, 3x3 kernels, and $\tanh$ activations. Adult uses an MLP model with two hidden layers of 256 units, while Dutch uses a logistic regression model. For all private methods, we use an RDP accountant (Mironov, 2017) with $\delta = 10^{-6}$. As a baseline, for DPSGD we set $\sigma = 1$, $C_0 = 0.5$ for Adult, $\sigma = 1$, $C_0 = 0.1$ for Dutch, and $\sigma = 0.8$, $C_0 = 1$ for image datasets. With this, training 20 epochs for tabular datasets, 60 epochs for MNIST and 30 epochs for CelebA gives $\epsilon = 3.41$ for Adult, $\epsilon = 2.27$ for Dutch, $\epsilon = 5.90$ for MNIST, and $\epsilon = 2.49$ for CelebA. DPSGD-F has negligibly

higher $\epsilon$, while our method achieves the same $\epsilon$ guarantees to three significant digits. Complete hyperparameters are given in App. B.2.

## 6.2 RESULTS

Table 2: Performance and Fairness metrics for MNIST

| METHOD | ACC 2 | ACC 8 | $\pi_2$ | $\pi_8$ | $\pi_{2,8}$ | LOSS 2 | LOSS 8 | $R_2$ | $R_8$ | $R_{2,8}$ |
|---|---|---|---|---|---|---|---|---|---|---|
| NON PRIVATE | 98.0±0.1 | 84.3±1.1 | - | - | - | 0.06±0.00 | 0.32±0.01 | - | - | - |
| DPSGD | 89.0±0.1 | 26.3±0.4 | 8.9±0.1 | 57.9±1.3 | 48.9±1.3 | 0.67±0.01 | 2.56±0.04 | 0.61±0.01 | 2.24±0.03 | 1.63±0.03 |
| DPSGD-F | 89.5±0.1 | 59.3±0.4 | 8.5±0.1 | 24.9±1.3 | 16.4±1.3 | 0.65±0.01 | 1.47±0.04 | 0.59±0.01 | 1.16±0.03 | 0.56±0.04 |
| DPSGD-G. | 90.6±0.2 | 62.0±2.6 | 7.4±0.1 | 22.2±2.6 | 14.8±2.7 | 0.34±0.01 | 1.31±0.04 | 0.28±0.01 | 0.99±0.03 | 0.71±0.04 |
| DPSGD-G.-A. | 92.0±0.2 | 65.5±1.2 | 6.0±0.2 | 18.8±0.9 | 12.8±0.8 | 0.35±0.01 | 1.20±0.04 | 0.29±0.01 | 0.89±0.03 | 0.60±0.03 |

Table 3: Performance and Fairness metrics for CelebA

| METHOD | ACC W/O | ACC W | $\pi_{W/O}$ | $\pi_W$ | $\pi_{W/O, W}$ | LOSS W/O | LOSS W | $R_{W/O}$ | $R_W$ | $R_{W/O, W}$ |
|---|---|---|---|---|---|---|---|---|---|---|
| NON PRIVATE | 95.8±0.1 | 89.7±0.4 | - | - | - | 0.11±0.00 | 0.24±0.01 | - | - | - |
| DPSGD | 86.5±0.2 | 74.0±0.6 | 9.3±0.3 | 15.7±0.6 | 6.4±0.7 | 0.60±0.01 | 1.34±0.05 | 0.49±0.01 | 1.10±0.05 | 0.61±0.05 |
| DPSGD-F | 91.8±0.2 | 79.7±0.5 | 4.0±0.2 | 10.0±0.6 | 6.0±0.6 | 0.32±0.01 | 0.97±0.04 | 0.21±0.01 | 0.73±0.04 | 0.52±0.04 |
| DPSGD-G. | 93.1±0.3 | 82.5±0.5 | 2.7±0.3 | 7.2±0.6 | 4.5±0.5 | 0.21±0.01 | 0.57±0.05 | 0.10±0.01 | 0.33±0.05 | 0.24±0.04 |
| DPSGD-G.-A. | 94.2±0.1 | 84.5±0.2 | 1.6±0.2 | 5.2±0.5 | 3.6±0.4 | 0.17±0.00 | 0.45±0.01 | 0.06±0.00 | 0.21±0.01 | 0.15±0.01 |

Tables 2 and 3 display the accuracy and loss, along with privacy cost and excessive risk metrics respectively for MNIST on classes 2 and 8 and CelebA on group W with eyeglasses, and group W/O without.[3] Recall that higher is better for accuracy, but for all other metrics lower is better. According to the one-sided Wilcoxon signed rank test, both global methods have statistically significant ($p < 0.05$) improvement over DPSGD on accuracy, loss, privacy cost gap, and excessive risk gap. Similarly, DPSGD-Global-Adapt has statistically significant improvement over DPSGD-Global and DPSGD-F on accuracy and loss. The same conclusions hold for the Adult dataset, and also for Dutch with the exception of DPSGD-Global being comparable to DPSGD in loss, see Tables 4 and 5 in App. B.7. We infer that the global scaling technique mitigates unfairness, while our modifications further improve utility.

Not only are final model metrics improved, we see that DPSGD-Global-Adapt trains more similarly to non-private SGD in Fig. 3 for Dutch (cf. Figs. 8, 9, and 10 in App. B.7 for Adult, MNIST, and CelebA). This shows the average train loss per iteration, and average norm of the batched gradient. The difference in loss for groups in DPSGD-Global-Adapt resembles that of the non-private method more closely than other methods. Consider Fig. 3 (bottom), where the group M average norm does not converge to 0 in DPSGD, a problem which is somewhat improved in DPSGD-F, while for FairLens the group F norms become much larger. In DPSGD-Global-Adapt the norms for both groups remain small, but importantly the gap between groups is reduced.

---

[3]The FairLens method (Tran et al., 2021a) is not compared for MNIST and CelebA because the author-provided code only handles binary classification problems, and does not scale to image datasets.

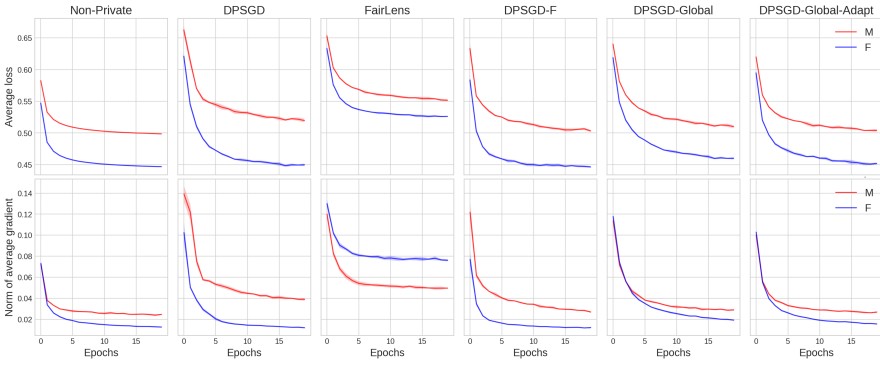

Figure 3: Dutch dataset. **Top**: Train loss per epoch. **Bottom**: $\|g_B\|$ averaged over batches per epoch.

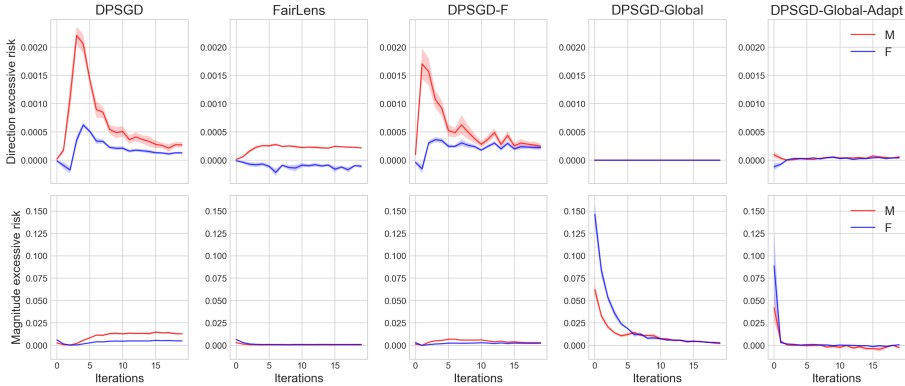

Figure 4: Adult dataset. **Top** $R_a^{\mathrm{dir}}$, excessive risk due to gradient misalignment per group. **Bottom** $R_a^{\mathrm{mag}}$, excessive risk due to magnitude error per group. See Prop. 2 for definitions.

Fig. 4 shows the excessive risk terms due to gradient misalignment $R_a^{\mathrm{dir}}$, and magnitude error $R_a^{\mathrm{mag}}$ for Adult at each iteration (see Figs. 11, 12, and 13 in App. B.7 for Dutch, MNIST, and CelebA). We see that global clipping almost completely removes direction errors as intended, but as a tradeoff increases magnitude error. However, we have argued that direction error is the more severe cause of disparate impact over the course of training, which is borne out by the results in Tables 1, 2 and 3, as well as 4, and 5 in App. B.7. Direction errors introduce bias which accumulates, whereas magnitude errors do not alter the convergence path, and noise errors add zero bias and tend to cancel out.

## 6.3 TIGHTNESS OF LOWER BOUNDS

In Fig. 5 we compare the usefulness of the lower bound of $R_a^{\mathrm{clip}} - R_b^{\mathrm{clip}}$ given in the proof of Theorem 3 in Tran et al. (2021a), to the lower bound we give in Prop. 3 for $R_a^{\mathrm{dir}} - R_b^{\mathrm{dir}}$. We see that while group 0 experiences disparate impact due to clipping, the lower bound from Tran et al. (2021a) is negative for each iteration, failing to capture that $R_0^{\mathrm{clip}} > R_1^{\mathrm{clip}}$. On the other hand, the true values of $R_0^{\mathrm{dir}} - R_1^{\mathrm{dir}}$ are closely lower-bounded in our version, such that disparate impact due to direction error is accurately predicted. The assumptions of Prop. 3 are discussed in App. B.6.

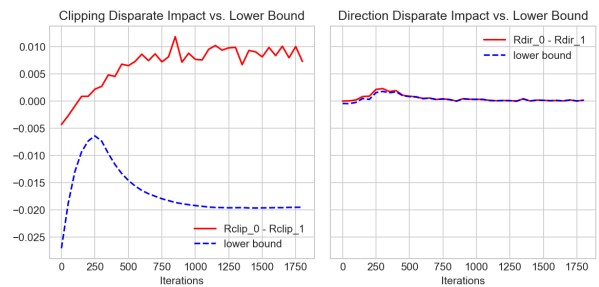

Figure 5: Comparison of excessive risk gaps $R_{0,1}$ to lower bounds on Adult. **Left**: $R_{0,1}$ due to clipping error, and bound from Tran et al. (2021a). **Right**: $R_{0,1}$ due to direction error, and bound from our Prop. 3.

## 7 DISCUSSION

In this paper we identified a core cause of disparate impact in DPSGD, gradient misalignment, and proposed a mitigating solution, global scaling. We empirically verified that global scaling is successful in improving fairness in terms of accuracy and loss over DPSGD and other fair baselines on several datasets. Our method has additional advantages over other fair baselines in that it does not require the collection of protected group data for training, does not involve disparate treatment, and it removes disparate impact for all groups simultaneously.

It is important to note that while global scaling is effective at reducing disparate impact by aligning gradients, it does not resolve the privacy-utility trade-off, which exists in any private mechanism fundamentally. Nor does it ensure that the model is non-discriminatory towards subgroups, only that adding privacy does not exacerbate unfairness. For example, biases in data collection or discriminatory modelling assumptions can cause disparate impact within the non-private model, which overlaying global scaling will not cure. Any models trained with global scaling should still be validated for fairness independently; failure to do so could unknowingly cause additional unfairness.

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

# A   THEORETICAL RESULTS

## A.1   PROOFS OF MAIN RESULTS

In this section we provide complete proofs for our theoretical contributions.

**Proposition 2.** *Consider the ERM problem with twice-differentiable loss $\ell$ with respect to the model parameters. The excessive risk due to clipping experienced by group $a \in [K]$ at iteration $t$ is approximated up to second order in $\|\theta_{t+1} - \theta_t\|$ as*

$$R_a^{\text{clip}} \approx \eta_t \left\langle g_{D_a}, \mathbb{E}\left[\left(1 - \frac{\|\bar{g}_B\|}{\|g_B\|}\right) g_B\right]\right\rangle + \frac{\eta_t^2}{2}\mathbb{E}\left[\left(\frac{\|\bar{g}_B\|^2}{\|g_B\|^2} - 1\right) g_B^T H_\ell^a g_B\right] \qquad (R_a^{\text{mag}})$$

$$+ \eta_t \left\langle g_{D_a}, \mathbb{E}\left[\frac{\|\bar{g}_B\|}{\|g_B\|}(g_B - M_B g_B)\right]\right\rangle + \frac{\eta_t^2}{2}\mathbb{E}\left[\frac{\|\bar{g}_B\|^2}{\|g_B\|^2}\left((M_B g_B)^T H_\ell^a (M_B g_B) - g_B^T H_\ell^a g_B\right)\right], \qquad (R_a^{\text{dir}})$$

*where $g_{D_a}$, $\bar{g}_{D_a}$ denote the average non-clipped and clipped gradients over group $a$ at iteration $t$, $H_\ell^a$ refers to the Hessian over group $a$, and $M_B$ is an orthogonal matrix such that $\bar{g}_B$ and $M_B g_B$ are colinear. The expectations are taken over batches of data.*

We remark that assuming a twice-differentiable loss is a mild requirement in machine learning where most loss functions and models are designed to be smooth enough for backpropagation.

*Proof.*

The proof is based on a Taylor expansion of the excessive risk, as in Tran et al. (2021a).

Let $M_B$ be an orthogonal matrix such that $\bar{g}_B = M_B\left(\frac{\|\bar{g}_B\|}{\|g_B\|}g_B\right)$. In this way, $\|\bar{g}_B\| = \left\|\frac{\|\bar{g}_B\|}{\|g_B\|}g_B\right\|$ and $g_B$ and $\frac{\|\bar{g}_B\|}{\|g_B\|}g_B$ are colinear, and so the former characterizes direction error, and the latter error in magnitude. The excessive risk due to error in magnitude for group $a$ at iteration $t$ is then given by

$$\mathbb{E}\left[\mathcal{L}\left(\theta_t - \eta_t \frac{\|\bar{g}_B\|}{\|g_B\|}g_B; D_a\right) - \mathcal{L}(\theta_t - \eta_t g_B; D_a)\right],$$

the cost in loss of using the update vector $\frac{\|\bar{g}_B\|}{\|g_B\|}g_B$ rather than $g_B$, where the expectation is over randomness of batch sampling. We perform second-order Taylor expansion of $\mathbb{E}\left[\mathcal{L}\left(\theta_t - \eta_t \frac{\|\bar{g}_B\|}{\|g_B\|}g_B; D_a\right)\right]$ and take the expectation to get that

$$\mathbb{E}\left[\mathcal{L}\left(\theta_t - \eta_t \frac{\|\bar{g}_B\|}{\|g_B\|}g_B; D_a\right)\right] \approx \mathcal{L}(\theta_t; D_a) - \eta_t \left\langle g_{D_a}, \mathbb{E}\left[\frac{\|\bar{g}_B\|}{\|g_B\|}g_B\right]\right\rangle + \frac{\eta_t^2}{2}\mathbb{E}\left[\frac{\|\bar{g}_B\|^2}{\|g_B\|^2}g_B^T H_\ell^a g_B\right].$$

Hence,

$$
\begin{aligned}
R_a^{\text{clip}} &= \eta_t \langle g_{D_a}, g_D - \bar{g}_D \rangle + \frac{\eta_t^2}{2} \left( \mathbb{E}[\bar{g}_B^T H_\ell^a \bar{g}_B] - \mathbb{E}[g_B^T H_\ell^a g_B] \right) \\
&= \eta_t \langle g_{D_a}, g_D - \bar{g}_D \rangle + \frac{\eta_t^2}{2} \left( \mathbb{E}[\bar{g}_B^T H_\ell^a \bar{g}_B] - \mathbb{E}[g_B^T H_\ell^a g_B] \right) \\
&\quad - \eta_t \left\langle g_{D_a}, \mathbb{E}\left[ \frac{\|\bar{g}_B\|}{\|g_B\|} g_B \right] \right\rangle + \frac{\eta_t^2}{2} \mathbb{E}\left[ \frac{\|\bar{g}_B\|^2}{\|g_B\|^2} g_B^T H_\ell^a g_B \right] \\
&\quad + \eta_t \left\langle g_{D_a}, \mathbb{E}\left[ \frac{\|\bar{g}_B\|}{\|g_B\|} g_B \right] \right\rangle - \frac{\eta_t^2}{2} \mathbb{E}\left[ \frac{\|\bar{g}_B\|^2}{\|g_B\|^2} g_B^T H_\ell^a g_B \right] \\
&= \eta_t \left\langle g_{D_a}, g_D - \mathbb{E}\left[ \frac{\|\bar{g}_B\|}{\|g_B\|} g_B \right] \right\rangle + \frac{\eta_t^2}{2} \left( \mathbb{E}\left[ \frac{\|\bar{g}_B\|^2}{\|g_B\|^2} g_B^T H_\ell^a g_B \right] - \mathbb{E}\left[ g_B^T H_\ell^a g_B \right] \right) \\
&\quad + \eta_t \left\langle g_{D_a}, \mathbb{E}\left[ \frac{\|\bar{g}_B\|}{\|g_B\|} g_B \right] - \bar{g}_D \right\rangle + \frac{\eta_t^2}{2} \left( \mathbb{E}\left[ \bar{g}_B^T H_\ell^a \bar{g}_B \right] - \mathbb{E}\left[ \frac{\|\bar{g}_B\|^2}{\|g_B\|^2} g_B^T H_\ell^a g_B \right] \right) \\
&= \eta_t \left\langle g_{D_a}, g_D - \mathbb{E}\left[ \frac{\|\bar{g}_B\|}{\|g_B\|} g_B \right] \right\rangle + \frac{\eta_t^2}{2} \mathbb{E}\left[ \left( \frac{\|\bar{g}_B\|^2}{\|g_B\|^2} - 1 \right) g_B^T H_\ell^a g_B \right] \qquad (R_a^{\text{mag}}) \\
&\quad + \eta_t \left\langle g_{D_a}, \mathbb{E}\left[ \frac{\|\bar{g}_B\|}{\|g_B\|} g_B \right] - \bar{g}_D \right\rangle + \frac{\eta_t^2}{2} \left( \mathbb{E}\left[ \bar{g}_B^T H_\ell^a \bar{g}_B \right] - \mathbb{E}\left[ \frac{\|\bar{g}_B\|^2}{\|g_B\|^2} g_B^T H_\ell^a g_B \right] \right). \\
&\qquad\qquad\qquad\qquad\qquad\qquad\qquad\qquad\qquad\qquad\qquad\qquad\qquad\qquad\qquad\qquad (R_a^{\text{dir}})
\end{aligned}
$$

We can also further simplify $R_a^{\text{dir}}$ by using that $\bar{g}_D = \mathbb{E}[\bar{g}_B]$, $\bar{g}_B = M_B \left( \frac{\|\bar{g}_B\|}{\|g_B\|} g_B \right)$ and that $M_B$ is a linear transformation

$$
\begin{aligned}
R_a^{\text{dir}} &= \eta_t \left\langle g_{D_a}, \mathbb{E}\left[ \frac{\|\bar{g}_B\|}{\|g_B\|} g_B - M_B \left( \frac{\|\bar{g}_B\|}{\|g_B\|} g_B \right) \right] \right\rangle \\
&\quad + \frac{\eta_t^2}{2} \left( \mathbb{E}\left[ \frac{\|\bar{g}_B\|^2}{\|g_B\|^2} (M_B g_B)^T H_\ell^a (M_B g_B) \right] - \mathbb{E}\left[ \frac{\|\bar{g}_B\|^2}{\|g_B\|^2} g_B^T H_\ell^a g_B \right] \right) \qquad (5) \\
&= \eta_t \left\langle g_{D_a}, \mathbb{E}\left[ \frac{\|\bar{g}_B\|}{\|g_B\|} (g_B - M_B g_B) \right] \right\rangle + \frac{\eta_t^2}{2} \mathbb{E}\left[ \frac{\|\bar{g}_B\|^2}{\|g_B\|^2} \left( (M_B g_B)^T H_\ell^a (M_B g_B) - g_B^T H_\ell^a g_B \right) \right].
\end{aligned}
$$

$\square$

**Proposition 3.** *Assume the loss $\ell$ is twice continuously differentiable and convex with respect to the model parameters. As well, assume that $\eta_t \leq (\max_{k \in [K]} \lambda_k)^{-1}$ where $\lambda_k$ is the maximum eigenvalue of the Hessian $H_\ell^k$. For groups $a, b \in [K]$, $R_a^{\text{dir}} > R_b^{\text{dir}}$ if*

$$
\mathbb{E}\left[ \|\bar{g}_B\| (\cos \theta_B^a - \cos \bar{\theta}_B^a) \right] > \frac{\|g_{D_b}\|}{\|g_{D_a}\|} \mathbb{E}\left[ \|\bar{g}_B\| (\cos \theta_B^b - \cos \bar{\theta}_B^b) \right] + \frac{\mathbb{E}[\|\bar{g}_B\|^2]}{\|g_{D_a}\|}, \qquad (6)
$$

*where $\theta_B^k = \angle(g_{D_k}, g_B)$ and $\bar{\theta}_B^k = \angle(g_{D_k}, \bar{g}_B)$ for a group $k \in [K]$. Furthermore, the bound is tight.*

Again, requiring a twice continuously differentiable loss is a mild requirement. However, when neural networks are used most loss functions are non-convex. Empirically we see in Fig. 5 that the lower bound can still apply in practice. The requirement on the learning rate is under the control of the practitioner, and we have verified that in practice it can be satisfied.

*Proof.*

This proof follows some steps presented in Lemma 2 of Tran et al. (2021a). We seek a simplified condition for when the following is positive,

$$
\begin{aligned}
R_a^{\text{dir}} - R_b^{\text{dir}} = \eta_t \left\langle g_{D_a}, \mathbb{E}\left[\frac{\|\bar{g}_B\|}{\|g_B\|}(g_B - M_B g_B)\right]\right\rangle - \eta_t \left\langle g_{D_b}, \mathbb{E}\left[\frac{\|\bar{g}_B\|}{\|g_B\|}(g_B - M_B g_B)\right]\right\rangle \\
+ \frac{\eta_t^2}{2}\mathbb{E}\left[\frac{\|\bar{g}_B\|^2}{\|g_B\|^2}\left((M_B g_B)^T H_\ell^a (M_B g_B) - g_B^T H_\ell^a g_B\right)\right] \\
- \frac{\eta_t^2}{2}\mathbb{E}\left[\frac{\|\bar{g}_B\|^2}{\|g_B\|^2}\left((M_B g_B)^T H_\ell^b (M_B g_B) - g_B^T H_\ell^b g_B\right)\right].
\end{aligned}
\tag{7}
$$

Looking at one of the inner product terms, we use that $\langle x, y\rangle = \|x\|\|y\|\cos(x,y)$ and linearity of expectation to obtain

$$
\begin{aligned}
\left\langle g_{D_a}, \mathbb{E}\left[\frac{\|\bar{g}_B\|}{\|g_B\|}(g_B - M_B g_B)\right]\right\rangle &= \mathbb{E}\left[\frac{\|\bar{g}_B\|}{\|g_B\|}\left(\langle g_{D_a}, g_B\rangle - \langle g_{D_a}, M_B g_B\rangle\right)\right] \\
&= \|g_{D_a}\|\mathbb{E}\left[\frac{\|\bar{g}_B\|}{\|g_B\|}\left(\|g_B\|\cos(g_{D_a}, g_B)\right.\right. \\
&\qquad\qquad\qquad \left.\left. - \|M_B g_B\|\cos(g_{D_a}, M_B g_B)\right)\right] \\
&= \|g_{D_a}\|\mathbb{E}\left[\|\bar{g}_B\|(\cos\theta_B^a - \cos\bar{\theta}_B^a)\right],
\end{aligned}
\tag{8}
$$

where $\theta_B^a = \angle(g_{D_a}, g_B)$ and $\bar{\theta}_B^a = \angle(g_{D_a}, M_B g_B) = \angle(g_{D_a}, \bar{g}_B)$. The last equality follows from the definition of $M_B$ such that $\bar{g}_B$ and $M_B g_B$ are aligned and $\|g_B\| = \|M_B g_B\|$.

We can also get a bound on the difference in conjugates of the Hessian, $\mathbb{E}\left[\frac{\|\bar{g}_B\|^2}{\|g_B\|^2}\left((M_B g_B)^T H_\ell^a (M_B g_B) - g_B^T H_\ell^a g_B\right)\right]$. Note that since we assume the loss $\ell$ is convex, the Hessian $H_\ell^a$ is positive semi-definite such that $x^T H_\ell^a x \geq 0$ for all vectors $x$. It follows that $\mathbb{E}[x^T H_\ell^a x] \geq 0$ and so using linearity of expectation,

$$
\mathbb{E}\left[\frac{\|\bar{g}_B\|^2}{\|g_B\|^2}\left((M_B g_B)^T H_\ell^a (M_B g_B) - g_B^T H_\ell^a g_B\right)\right] \leq \mathbb{E}\left[\frac{\|\bar{g}_B\|^2}{\|g_B\|^2}(M_B g_B)^T H_\ell^a (M_B g_B)\right].
\tag{9}
$$

Since $\ell$ is twice continuously differentiable we have that $H_\ell^a$ is symmetric and hence $x^T H_\ell^a x \leq \lambda_a\|x\|^2$ where $\lambda_a$ is the maximum eigenvalue of $H_\ell^a$. We then again use that $\|M_B g_B\| = \|g_B\|$ and linearity of expectation to obtain

$$
\mathbb{E}\left[\frac{\|\bar{g}_B\|^2}{\|g_B\|^2}\left((M_B g_B)^T H_\ell^a (M_B g_B) - g_B^T H_\ell^a g_B\right)\right] \leq \lambda_a \mathbb{E}\left[\|\bar{g}_B\|^2\right].
\tag{10}
$$

Similar analysis gives that $\mathbb{E}\left[\frac{\|\bar{g}_B\|^2}{\|g_B\|^2}\left((M_B g_B)^T H_\ell^a (M_B g_B) - g_B^T H_\ell^a g_B\right)\right] \geq -\lambda_a \mathbb{E}[\|\bar{g}_B\|^2]$.

Combining the above, it follows that

$$
\begin{aligned}
R_a^{\text{dir}} - R_b^{\text{dir}} \geq \eta_t\left(\|g_{D_a}\|\mathbb{E}\left[\|\bar{g}_B\|(\cos\theta_B^a - \cos\bar{\theta}_B^a)\right] - \|g_{D_b}\|\mathbb{E}\left[\|\bar{g}_B\|(\cos\theta_B^b - \cos\bar{\theta}_B^b)\right]\right) \\
- \frac{\eta_t^2}{2}(\lambda_a + \lambda_b)\mathbb{E}[\|\bar{g}_B\|^2],
\end{aligned}
\tag{11}
$$

and since we assume $\eta_t \leq \frac{1}{\max_{k\in[K]}\lambda_k}$,

$$
\begin{aligned}
R_a^{\text{dir}} - R_b^{\text{dir}} \geq \eta_t\left(\|g_{D_a}\|\mathbb{E}\left[\|\bar{g}_B\|(\cos\theta_B^a - \cos\bar{\theta}_B^a)\right]\right. \\
\left. - \|g_{D_b}\|\mathbb{E}\left[\|\bar{g}_B\|(\cos\theta_B^b - \cos\bar{\theta}_B^b)\right] - \mathbb{E}[\|\bar{g}_B\|^2]\right).
\end{aligned}
\tag{12}
$$

It follows that $R_a^{\text{dir}} > R_b^{\text{dir}}$ when the following is satisfied:

$$
\mathbb{E}\left[\|\bar{g}_B\|(\cos\theta_B^a - \cos\bar{\theta}_B^a)\right] > \frac{\|g_{D_b}\|}{\|g_{D_a}\|}\mathbb{E}\left[\|\bar{g}_B\|(\cos\theta_B^b - \cos\bar{\theta}_B^b)\right] + \frac{\mathbb{E}[\|\bar{g}_B\|^2]}{\|g_{D_a}\|}.
\tag{13}
$$

Finally, to see that the bound is tight we simply note that the inequalities that were introduced can all be saturated simultaneously. In Eq. 9 we require that $g_B^T H_\ell^a g_B = 0$, and in Eq. 10 we require $(M_B g_B)^T H_\ell^a (M_B g_B) = \lambda_a \|M_B g_B\|^2$ for each batch. These independent conditions can plausibly be met for some $H_\ell^a$, $g_B$, and $M_B$. The only other inequality introduced is the assumption $\eta_t \leq \frac{1}{\max_{k \in [K]} \lambda_k}$, which we can strengthen to $\eta_t = \frac{1}{\max_{k \in [K]} \lambda_k}$ for the sake of achieving saturation.

$\square$

## A.2 ALTERNATE DECOMPOSITIONS OF THE CLIPPING ERROR

In Sec. A.2 we proposed a decomposition of the clipped batch gradient into parts representing magnitude and direction error, $\bar{g}_B = M_B \left( \frac{\|\bar{g}_B\|}{\|g_B\|} g_B \right)$. We presented a simple experiment in Table 1 to demonstrate that direction error causes the most severe problems for the final performance of models, and analysed the contributions of the two effects to the excessive risk in Prop. 2.

However, the decomposition we used is not unique, and furthermore it is not possible to completely isolate the two effects in the excessive risk analysis. For example, if we think of magnitude error as the difference in loss between using update vector $g_B$ and $\frac{\|\bar{g}_B\|}{\|g_B\|} g_B$ ($\gamma$ in Fig. 6), then it follows that the remaining error is due to gradient misalignment, in other words, the difference in loss between using update vector $\frac{\|\bar{g}_B\|}{\|g_B\|} g_B$ and $\bar{g}_B$ ($\lambda$ in Fig. 6). In this example, the error due to gradient misalignment includes both error in direction and error in magnitude, while magnitude error is "pure",

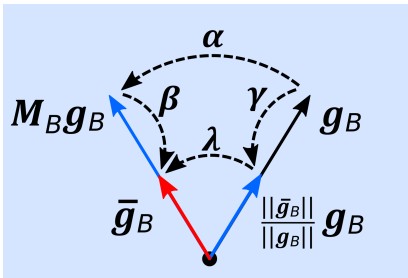

Figure 6: Decomposition of steps between $g_B$ and $\bar{g}_B$.

$$R_a^{\text{mag}} = \mathbb{E}[\mathcal{L}(\theta_t - \eta_t \frac{\|\bar{g}_B\|}{\|g_B\|} g_B; D_a) - \mathcal{L}(\theta_t - \eta_t g_B; D_a)], \tag{14}$$

$$R_a^{\text{dir}} = \mathbb{E}[\mathcal{L}(\theta_t - \eta_t \bar{g}_B; D_a) - \mathcal{L}(\theta_t - \eta_t \frac{\|\bar{g}_B\|}{\|g_B\|} g_B; D_a)]. \tag{15}$$

A different way of decomposing the clipping error is considering the direction error as the difference in loss between using update vector $g_B$ and $M_B g_B$ ($\alpha$ in Fig. 6). In this case, direction error is pure, i.e. does not include difference in magnitudes. It follows that the remaining error is magnitude error, so is the difference in loss between using update vector $M_B g_B$ and $\bar{g}_B$ ($\beta$ in Fig. 6). Thus, the magnitude error in this case quantifies the difference in loss of scaling the already misaligned $\bar{g}_B$,

$$R_a^{\text{dir}*} = \mathbb{E}[\mathcal{L}(\theta_t - \eta_t M_B g_B; D_a) - \mathcal{L}(\theta_t - \eta_t g_B; D_a)], \tag{16}$$

$$R_a^{\text{mag}*} = \mathbb{E}[\mathcal{L}(\theta_t - \eta_t \bar{g}_B; D_a) - \mathcal{L}(\theta_t - \eta_t M_B g_B; D_a)]. \tag{17}$$

In our analysis we used the first decomposition where magnitude error can be completely corrected by an adjustment of the learning rate, and direction error, what we hypothesized to be the largest cause of disparate impact, is the remaining part of the clipping error. For completeness, by using the second decomposition we can derive alternative versions of Props. 2 and 3:

**Proposition 2\*.** *Consider the ERM problem with twice-differentiable loss $\ell$ with respect to the model parameters. The excessive risk due to clipping experienced by group $a \in [K]$ at iteration $t$ is approximated up to second order in $\|\theta_{t+1} - \theta_t\|$ as*

$$R_a^{\text{clip}} \approx \eta_t \left\langle g_{D_a}, \mathbb{E}\left[ \left( \frac{\|g_B\|}{\|\bar{g}_B\|} - 1 \right) \bar{g}_B \right] \right\rangle + \frac{\eta_t^2}{2} \mathbb{E}\left[ \left( 1 - \frac{\|g_B\|^2}{\|\bar{g}_B\|^2} \right) \bar{g}_B^T H_\ell^a \bar{g}_B \right], \tag{$R_a^{\text{mag}*}$}$$

$$+ \mathbb{E}\left[ \eta_t \langle g_{D_a}, g_D - M_B g_B \rangle \right] + \frac{\eta_t^2}{2} \mathbb{E}\left[ (M_B g_B)^T H_\ell^a (M_B g_B) - g_B^T H_\ell^a g_B \right], \tag{$R_a^{\text{dir}*}$}$$

*where $g_{D_a}$, $\bar{g}_{D_a}$ denote the average non-clipped and clipped gradients over group $a$ at iteration $t$, $H_\ell^a$ refers to the Hessian over group $a$, and $M_B$ is an orthogonal matrix such that $\bar{g}_B$ and $M_B g_B$ are colinear. The expectations are taken over batches of data.*

**Proposition 3\*.** *Assume the loss $\ell$ is twice continuously differentiable and convex with respect to the model parameters. As well, assume that $\eta_t \leq (\max_{k \in [K]} \lambda_k)^{-1}$ where $\lambda_k$ is the maximum*

*eigenvalue of the Hessian $H_\ell^k$. For groups $a, b \in [K]$, $R_a^{\mathrm{dir}} > R_b^{\mathrm{dir}}$ if*

$$\mathbb{E}\left[\|g_B\|(\cos\theta_B^a - \cos\bar{\theta}_B^a)\right] > \frac{\|g_{D_b}\|}{\|g_{D_a}\|}\mathbb{E}\left[\|g_B\|(\cos\theta_B^b - \cos\bar{\theta}_B^b)\right] + \frac{\mathbb{E}[\|g_B\|]}{\|g_{D_a}\|} \qquad (18)$$

*where $\theta_B^k = \angle(g_{D_k}, g_B)$ and $\bar{\theta}_B^k = \angle(g_{D_k}, \bar{g}_B)$ for a group $k \in [K]$.*

We omit the proofs since they are directly analogous to those in App. A.1.

# B  EXPERIMENTAL DETAILS

## B.1  DATASET PREPROCESSING

**MNIST**  We use the artificially unbalanced MNIST training dataset where class 8 is sampled with probability 9% such that class 8 only constitutes about 1% of the dataset on average. This gives about 6000 data samples for each class, other than class 8 with about 500. The protected group values are the class labels. As in Xu et al. (2021), we compare models on how they treat the under-represented class 8 versus the well-represented class 2. The test set remains balanced, with approximately 1000 samples for each class. Data is scaled to be in the domain [0,1].

**Adult**  The original Adult dataset[4] consists of 48,842 samples, reduced to 45,222 by removing all samples with missing values. The "final weight" feature is removed and the "race" attribute is discretized by {white, non-white}, giving 5 numerical, 3 binary and 6 categorical features. The numerical features are normalized and the categorical features are one-hot encoded. As is typical in the fairness literature, choices for the protected attribute are "sex", "race" (binary) and possibly the discretized "age". We use "sex" by default. The classification label is "income" (whether or not income exceeds $50,000). Prior to sampling, the Adult dataset is unbalanced with respect to sex with 30,527 males and 14,695 females. We sample a balanced dataset as in Xu et al. (2021) with 14,000 females and 14,000 males on average.

**Dutch**  The Dutch dataset van der Laan (2000)[5] is preprocessed by dropping underage samples (14 and under) and removing the "weight" feature. As well, all "unemployed" samples are removed, as well as those with missing or middle-level "occupation", for a total of 60,420 samples. Specifically, "occupation" values 3,6,7,8 are considered middle-level. "Occupation" is then made binary by considering values 4,5,9 as low-level professions (0) and 1,2 as high-level professions (1). The binary classification task is to predict "occupation", given the rest of the features. We consider "sex" as the protected group attribute. The processed dataset is balanced with respect to "sex" with 30,147 male and 30,273 female samples.

We use an 80/20 train/test split for both tabular datasets.

**CelebA**  The CelebA dataset (Liu et al., 2015)[6], consists of 64x64 pixel RGB images of celebrity faces, along with binary attributes describing each image. Many of these attributes are subjective, but we chose to use the most objective ones for training and group labels. We used the binary attribute "Male" for the classification target, which is roughly balanced at 84,434 males in 202,599 total images. The attribute "Eyeglasses" was our protected group label; although wearing eyeglasses in public typically does not construe sensitive information, we used this attribute because it was objectively defined, and formed a minority group which was empirically more difficult for models to classify accurately. Of the male images, 10,478 have eyeglasses, while only 2,715 female images have them. The training/validation/test split is provided with the dataset and is roughly in a 80/10/10 ratio.

---

[4]The Adult dataset is available at archive.ics.uci.edu/ml/datasets/Adult.

[5]The Dutch dataset is also available through the work of Le Quy et al. (2022) at raw.githubusercontent.com/tailequy/fairness_dataset/main/Dutch_census/dutch_census_2001.arff.

[6]We accessed this dataset via kaggle.com/datasets/jessicali9530/celeba-dataset.

## B.2 EXPERIMENT SETTINGS

We set $\sigma = 1$, $C_0 = 0.5$ for Adult, $\sigma = 1$, $C_0 = 0.1$ for Dutch, while for MNIST and CelebA, we set $\sigma = 0.8$ and $C_0 = 1$. For DPSGD-F, the gradient noise is unchanged $\sigma_2 = \sigma$, and $\sigma_1 = 10\sigma_2$. For FairLens, we use regularization weights as in Tran et al. (2021a), $\lambda_1 = \lambda_2 = 1$. For non-global methods, the learning rate is $\eta_t = 0.01$ for all iterations $t$ and all datasets except Dutch which has $\eta_t = 0.8$. For DPSGD-Global we have $\eta_t = 1$, $Z = 50$ for Adult, $\eta_t = 2$, $Z = 1$ for Dutch, $\eta_t = 0.2$, $Z = 100$ for MNIST, and $\eta_t = 0.1$, $Z = 100$ for CelebA. For DPSGD-Global-Adapt we have $\sigma_2 = 10, Z = 50, \eta_Z = 0.1$ for all datasets (the only exception is for CelebA $Z = 100$), $\eta_t = 0.2, \tau = 1$ for Adult, $\eta = 1, \tau = 1$ for Dutch, and $\eta = 0.1, \tau = 0.7$ for MNIST and CelebA. All methods for all datasets use training and test batches of size 256.

Experiments were conducted on single TITAN V GPU machines. Approximately four GPU-days were used to train all methods over five seeds for the four datasets.

## B.3 IMPLEMENTATION DETAILS

The excessive risk terms for different groups ($R_a^{\text{clip}}$ and $R_a^{\text{noise}}$ in Prop. 1 and $R_a^{\text{mag}}$ and $R_a^{\text{dir}}$ in Prop. 2) all involve the Hessian of the loss function with respect to the model parameters. Calculating the Hessian as a matrix is computationally expensive, but more crucially requires memory that scales quadratically in the number of parameters. In the previous work studying $R_a^{\text{clip}}$ and $R_a^{\text{noise}}$, Tran et al. (2021a) use the PyHessian library to compute the Hessian as a matrix, and then used it to compute the products and traces needed for $R_a^{\text{clip}}$ and $R_a^{\text{noise}}$. Because this approach incurs a high memory burden, the models trained were limited to small MLPs with a single hidden layer of 20 hidden units.[7]

In our implementation, provided as supplemental material, we avoid computing the Hessian as a matrix altogether which allows us to scale our experiments to common image datasets. For the four datasets, our models have parameter counts of $N = 91650$ for Adult, $N = 120$ for Dutch, $N = 80522$ for MNIST, and $N = 120722$ for CelebA, which would produce Hessian matrices with up to 14.5 billion entries. Instead, we compute the terms involving Hessians like $H_\ell^a g_B$ through Hessian-vector products (HVPs) using the functorch[8] library with PyTorch 1.11. Using HVPs requires memory comparable to that used when computing gradients for SGD.

For the trace of the Hessian matrix, also called the Laplacian, one possible approach that does not require realizing the entire matrix in memory is to compute HVPs with unit vectors to isolate each diagonal element: $\text{Tr}(H_\ell^a) = \sum_{i=1}^{N} I_i^T H_\ell^a I_i$ where $I_i$ is the $i$th column of the identity matrix. While exact, this approach requires $N$ HVPs for each group $a \in K$, of which there are at least two. Since this method is much too expensive for even the simple MLPs and CNNs we used, we instead employed Hutchinson's trace estimator (Hutchinson, 1990) to estimate $\text{Tr}(H_\ell^a) = \mathbb{E}_z[z^T H_\ell^a z]$. This estimator is unbiased when $z$ is drawn from a Rademacher distribution which we used, and only requires $n$ HVPs per group, where $n$ can be chosen as large as required for convergence of the estimate. In practice we used $n = 100$.

Additionally, whereas Tran et al. (2021a) replace dataset gradients $g_D$ and $g_{D_a}$ with batch gradients when computing $R_a^{\text{clip}}$ and $R_a^{\text{noise}}$ in Prop. 1, we use the exact $g_D$ and $g_{D_a}$. This eliminates an easily preventable source of noise in our results.

To further reduce computation time, we only evaluate excessive risk terms (Hessians) every 50, 100, 200, or 200 iterations for the Adult, Dutch, MNIST, and CelebA datasets respectively.

## B.4 DIRECTION ERROR IS MORE SEVERE THAN MAGNITUDE ERROR

As noted earlier, Prop. 2 only evaluates excessive risk for a single iteration, not necessarily capturing how each of $R_a^{\text{dir}}$ and $R_a^{\text{mag}}$ contribute to convergence and disparate impact over the course of training. In order to evaluate the full impact of magnitude error and error due to gradient misalignment, we consider the difference in final loss and accuracy between models which have zero magnitude

---

[7]See implementation available at openreview.net/forum?id=7EFdodSWee4.

[8]See documentation at pytorch.org/functorch/stable/.

error and zero direction error in Table 1. In these experiments, we consider zero magnitude error to be when $\|\bar{g}_B\| = \|g_B\|$ for all batches, and zero direction error to be when $g_B$ and $\bar{g}_B$ are aligned for all batches. Note that these definitions correspond to comparing update vectors $g_B$ and $\frac{\|g_B\|}{\|\bar{g}_B\|}\bar{g}_B$ for the zero magnitude error experiment, and comparing update vectors $g_B$ and $\frac{\|\bar{g}_B\|}{\|g_B\|}g_B$ for the zero direction error experiment. These do not correspond to the definitions of $R_a^{\text{dir}}$ and $R_a^{\text{mag}}$ in Prop. 2, but capture the intuitive definitions of direction and magnitude error. As described in App. A.2, while $R_a^{\text{clip}} = R_a^{\text{mag}} + R_a^{\text{dir}}$, direction error and magnitude error cannot be purely separated with any definition of $R_a^{\text{mag}}, R_a^{\text{dir}}$.

## B.5 BASELINE METHODS

We compared our approach DPSGD-Global-Adapt with its predecessor DPSGD-Global, which was designed to improve convergence, not fairness, as well as two approaches specifically designed to improve fairness.

DPSGD-Global (Bu et al., 2021) is presented in Alg. 2, and involves scaling almost all per-sample gradients by a global factor rather than only scaling large gradients with $\|g_i\| > C_0$ by a norm-dependent factor. We say "almost all", because scaling alone does not provide a strict upper bound on the sensitivity, as required for an application of the Gaussian mechanism, see Fig. 2b. The method additionally clips gradients to zero if their norm is above a strict upper bound $Z$, which we found to be unnecessarily aggressive. Otherwise, the global scaling factor is $C_0/Z$, which ensures that the sensitivity, namely $C_0$, is finite. The advantage of DPSGD-Global is that it can better preserve the direction of $\bar{g}_B$, especially when no gradients are clipped to zero. Hence, Bu et al. (2021) advocate for setting $Z$ larger than $\|g_i\|$ for any sample in the batch. The drawback of a large $Z$ is that all gradients are scaled down by a larger factor, so the convergence will be slowed unless the learning rate is increased to compensate. Setting $Z$ is itself a challenge because we cannot inspect the batch to determine $\max_i \|g_i\|$ without accounting for that expense in our privacy budget. In Sec. 5 we described how DPSGD-Global-Adapt resolves these concerns, first by clipping less aggressively, to $C_0$ instead of 0, while maintaining the same sensitivity, and second by adaptively setting $Z$ each round according to a private estimate of how many gradients in a batch exceeded $\tau \cdot Z$ (using the tolerance threshold $\tau$).

Xu et al. (2021) designed DPSGD-F as a method for removing disparate impact caused by DPSGD by adaptively setting the clipping threshold for different protected groups. The method was based on the observation that negatively impacted groups tended to have large gradient norms which were affected more by clipping. Hence, the clipping threshold is raised for groups with larger gradient norms, based on a private estimate of how many gradients per-group have $\|g_i\| > C_0$. Given large enough batch sizes, the private estimate can be done with much more noise as compared to the gradient update, so it does not meaningfully increase the privacy budget.

One drawback of this approach is that it requires group label information for every datapoint in the training set. In practice, especially in highly regulated industries, such information may not be permissible to use or even collect. Collecting additional private information from data subjects on protected attributes can itself be a negative process and creates unnecessary privacy risks. One major advantage of DPSGD-Global-Adapt is that it reduces unfairness without ever using group label information.

While each group is clipped using its own threshold, noise is added to the batched gradient based on the sensitivity, determined by the largest group threshold. While all groups receive the same theoretical privacy guarantee in terms of $(\epsilon, \delta)$, groups that are clipped to smaller thresholds may enjoy stronger empirical privacy guarantees, as determined for example by adversarial attacks (Jagielski et al., 2020; Nasr et al., 2021). Hence, it appears likely that DPSGD-F can produce unfairness in the amount of privacy afforded to different groups.

DPSGD-F is shown in Alg. 3. Note that we present the algorithm as implemented in the author's codebase, not as written in their paper. In our experiments we use the version shown in Alg. 3.

Our final baseline, referred to as "FairLens" was developed in (Tran et al., 2021a) to reduce excessive risk from clipping, $R_a^{\text{clip}}$, and adding noise, $R_a^{\text{noise}}$. Regularization terms are added to the loss function in DPSGD that specifically target these sources of excessive risk. The source of $R^{\text{noise}}$ was identified

---

**Algorithm 3** DPSGD-F

---

**Require:** Iterations $T$, Dataset $D$, sampling rate $q$, clipping bound $C_0$, noise multipliers $\sigma_1, \sigma_2$,
       learning rates $\eta_t$

   Initialize $\theta_0$ randomly
   **for** $t$ in $0, \ldots, T-1$ **do**
     $B \leftarrow$ Poisson sample of $D$ with rate $q$
     **for** $(x_i, a_i, y_i)$ in $B$ **do**
       $g_i \leftarrow \nabla_\theta \ell(f_{\theta_t}(x_i), y_i)$           ▷ *Compute per-sample gradients*
     **for** $k$ in $[K]$ **do**
       $m^k \leftarrow \left| \{ i : \|g_i^k\| > C_0 \} \right|$      ▷ *Count samples per-group above/below clipping bound*
       $o^k \leftarrow \left| \{ i : \|g_i^k\| \le C_0 \} \right|$
     $\{\tilde{m}^k, \tilde{o}^k\}_{k \in [K]} \leftarrow \{m^k, o^k\}_{k \in [K]} + \mathcal{N}(0, \sigma_1^2 \mathbb{I})$     ▷ *Privatize unit sensitivity count vectors*
     $\{\tilde{m}^k, \tilde{o}^k\}_{k \in [K]} \leftarrow \{\max(\lfloor \tilde{m}^k \rfloor, 0), \max(\lfloor \tilde{o}^k \rfloor, 0)\}_{k \in [K]}$       ▷ *Postprocessing*
     $\tilde{m} = \sum_{k \in [K]} \tilde{m}^k$
     **for** $k$ in $[K]$ **do**
       $\tilde{b}^k = \tilde{m}^k + \tilde{o}^k$
       $C_k = C_0 \cdot \left( 1 + \frac{\tilde{m}^k / \tilde{b}^k}{\tilde{m} / |B|} \right)$
     **for** $(x_i, a_i, y_i)$ in $B$ **do**
       $\bar{g}_i \leftarrow g_i \cdot \min\left(1, \frac{C_k}{\|g_i\|}\right)$ where $k = a_i$     ▷ *Clip according to per-group clipping bounds*
     $\tilde{g}_B \leftarrow \frac{1}{|B|} \left( \sum_{i \in B} \bar{g}_i + \mathcal{N}(0, \sigma_2^2 C_0^2 \mathbb{I}) \right)$
     $\theta_{t+1} \leftarrow \theta_t - \eta_t \tilde{g}_B$

---

to involve the per-group Laplacian of the loss $\ell$ with respect to model parameters - a second order derivative whose computation scales poorly with model size. To avoid this difficulty, the authors used a stand-in for the Laplacian based on the distance of a point to the decision boundary.

Our implementation is directly based off of code made available by the authors on OpenReview at openreview.net/forum?id=7EFdodSWee4. The version implemented in their code is shown in Alg. 4, and assumes there are only two mutually exclusive protected groups, denoted $a$ and $b$. Hence, it is not applicable to the MNIST dataset. We also attempted to use this code for our CelebA experiments but found that the implementation did not scale to the simple CNNs we used. Therefore, we omitted FairLens from the CelebA experiments.

## B.6   Verifying Assumptions on Lower Bound

In Sec. 6.3 and Fig. 5 we compared the usefulness of our lower bound on $R_a^{\text{dir}} - R_b^{\text{dir}}$ from Prop. 3, to a previous lower bound in the literature. For our lower bound to be valid, the assumptions of Prop. 3 should be satisfied. The first assumption, that the loss is twice continuously differentiable with respect to the model parameters, holds since the model architecture is an MLP with $\tanh$ activations. However, the loss is not in general convex. The third assumption, that the inverse of the learning rate upper bounds the largest eigenvalue of any group's Hessian, is checked empirically for each iteration in Fig. 7.

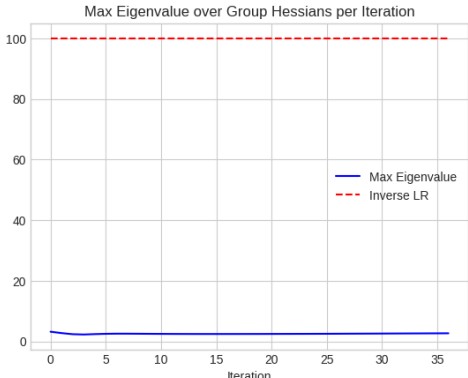

Figure 7: The maximum eigenvalue of any group's Hessian remains below $\eta_t^{-1}$.

---

**Algorithm 4** FairLens

---

**Require:** Iterations $T$, Dataset $D$, sampling rate $q$, clipping bound $C_0$, noise multiplier $\sigma$, learning rates $\eta_t$, regularization weights $\gamma_1, \gamma_2$

Initialize $\theta_0$ randomly

**for** $t$ in $0, \ldots, T-1$ **do**

$\quad B \leftarrow$ Poisson sample of $D$ with rate $q$

$\quad$**for** $(x_i, a_i, y_i)$ in $B$ **do**

$\quad\quad g_i \leftarrow \nabla_\theta \ell(f_{\theta_t}(x_i), y_i)$ $\qquad\qquad\qquad\qquad$ ▷ *Compute per-sample gradients of original loss*

$\quad\quad \bar{g}_i \leftarrow g_i \cdot \min\left(1, \frac{C_0}{\|g_i\|}\right)$

$\quad g_B \leftarrow \frac{1}{|B|} \sum_{i \in B} g_i$

$\quad \bar{g}_B \leftarrow \frac{1}{|B|} \sum_{i \in B} \bar{g}_i$

$\quad$**for** $k$ in $\{a, b\}$ **do**

$\quad\quad g_{B_k} \leftarrow \frac{1}{|B_k|} \sum_{i \in B, a_i = k} g_i$

$\quad\quad f_k \leftarrow \frac{1}{|B_k|} \sum_{i \in B, a_i = k} f_{\theta_t}(x_i)$

$\quad R_1 = |\langle g_{B_a} - g_{B_b}, \bar{g}_B - g_B \rangle|$

$\quad R_2 = \frac{1}{2}(f_a \cdot (1 - f_a) + f_b \cdot (1 - f_b))$

$\quad \mathcal{L} = \ell(f_{\theta_t}(x_i), y_i) + \gamma_1 R_1 + \gamma_2 R_2$ $\qquad\qquad\qquad$ ▷ *Define regularized loss*

$\quad$**for** $(x_i, a_i, y_i)$ in $B$ **do**

$\quad\quad g'_i \leftarrow \nabla_\theta \mathcal{L}(f_{\theta_t}(x_i), y_i)$ $\qquad$ ▷ *Compute per-sample gradients of regularized loss*

$\quad\quad \bar{g}'_i \leftarrow g'_i \cdot \min\left(1, \frac{C_0}{\|g'_i\|}\right)$ $\qquad\qquad\qquad\qquad$ ▷ *Clip to ensure finite sensitivity*

$\quad \tilde{g}'_B \leftarrow \frac{1}{|B|} \left( \sum_{i \in B} \bar{g}'_i + \mathcal{N}(0, \sigma^2 C_0^2 \mathbb{I}) \right)$

$\quad \theta_{t+1} \leftarrow \theta_t - \eta_t \tilde{g}'_B$

---

## B.7 ADDITIONAL RESULTS

In this section we complete the set of experimental results shown in Sec. 6 over all datasets and methods. All results are averaged over five random seeds with one standard error shown.

Table 4: Performance and Fairness metrics for Adult dataset

| METHOD | ACC M | ACC F | $\pi_M$ | $\pi_F$ | $\pi_{M,F}$ | LOSS M | LOSS F | $R_M$ | $R_F$ | $R_{M,F}$ |
|---|---|---|---|---|---|---|---|---|---|---|
| NON PRIVATE | 80.5±0.4 | 92.2±0.1 | - | - | - | 0.40±0.00 | 0.19±0.00 | - | - | - |
| DPSGD | 69.9±0.4 | 88.5±0.1 | 10.6±0.3 | 3.6±0.1 | 6.9±0.3 | 0.78±0.01 | 0.40±0.01 | 0.39±0.00 | 0.21±0.01 | 0.17±0.01 |
| FAIRLENS | 68.8±0.4 | 88.5±0.1 | 11.7±0.2 | 3.7±0.1 | 7.9±0.2 | 0.57±0.00 | 0.42±0.00 | 0.18±0.00 | 0.23±0.00 | 0.05±0.00 |
| DPSGD-F | 78.0±0.6 | 89.4±0.1 | 2.5±0.3 | 2.7±0.1 | 0.2±0.3 | 0.49±0.00 | 0.31±0.01 | 0.09±0.00 | 0.12±0.00 | 0.02±0.01 |
| DPSGD-G. | 78.5±0.5 | 89.9±0.1 | 2.0±0.2 | 2.2±0.1 | 0.2±0.2 | 0.43±0.00 | 0.25±0.00 | 0.04±0.00 | 0.05±0.00 | 0.02±0.00 |
| DPSGD-G.-A. | 80.7±0.4 | 92.3±0.1 | −0.1±0.1 | −0.1±0.1 | 0.0±0.1 | 0.39±0.00 | 0.18±0.00 | 0.00±0.00 | 0.00±0.00 | 0.00±0.00 |

Table 5: Performance and Fairness metrics for Dutch dataset

| METHOD | ACC M | ACC F | $\pi_M$ | $\pi_F$ | $\pi_{M,F}$ | LOSS M | LOSS F | $R_M$ | $R_F$ | $R_{M,F}$ |
|---|---|---|---|---|---|---|---|---|---|---|
| NON PRIVATE | 79.9±0.2 | 86.9±0.0 | - | - | - | 0.499±0.000 | 0.447±0.000 | - | - | - |
| DPSGD | 76.0±0.2 | 86.4±0.1 | 3.8±0.3 | 0.4±0.0 | 3.4±0.4 | 0.520±0.001 | 0.450±0.001 | 0.021±0.001 | 0.003±0.001 | 0.018±0.002 |
| FAIRLENS | 78.6±0.3 | 86.9±0.1 | 1.3±0.2 | −0.1±0.0 | 1.4±0.2 | 0.552±0.000 | 0.526±0.000 | 0.053±0.000 | 0.079±0.000 | 0.026±0.001 |
| DPSGD-F | 78.9±0.2 | 86.6±0.1 | 0.9±0.1 | 0.3±0.0 | 0.7±0.1 | 0.503±0.001 | 0.447±0.001 | 0.005±0.001 | 0.000±0.001 | 0.005±0.001 |
| DPSGD-G. | 79.0±0.2 | 86.5±0.1 | 0.8±0.1 | 0.4±0.0 | 0.4±0.2 | 0.510±0.001 | 0.460±0.001 | 0.012±0.001 | 0.013±0.001 | 0.002±0.001 |
| DPSGD-G.-A. | 79.4±0.1 | 86.7±0.1 | 0.4±0.2 | 0.2±0.0 | 0.2±0.2 | 0.504±0.001 | 0.452±0.001 | 0.006±0.001 | 0.005±0.001 | 0.001±0.001 |

First we look at the final performance and fairness metrics on the test set for Adult in Table 4 and Dutch in Table 5 (cf. MNIST in Table 2 and CelebA in Table 3). We see that FairLens is inconsistent in reducing the privacy cost gap and excessive risk gap compared to DPSGD. DPSGD-F improves both fairness metrics while achieving better performance. DPSGD-Global improves over or is comparable to DPSGD-F in all metrics, and does so without requiring access to protected group membership information. Our method DPSGD-Global-Adapt further improves both performance and fairness by clipping less aggressively and adaptively setting the upper clipping threshold $Z$.

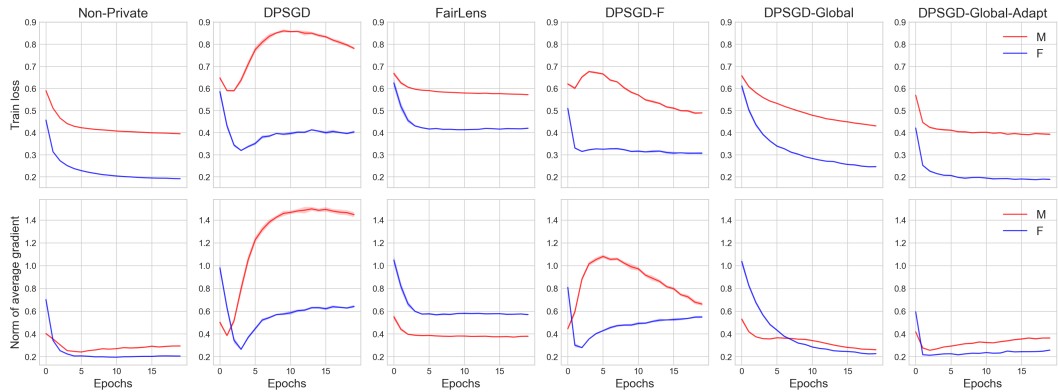

Figure 8: Adult dataset. **Top**: Train loss per epoch. **Bottom**: $\|g_B\|$ averaged over batches per epoch.

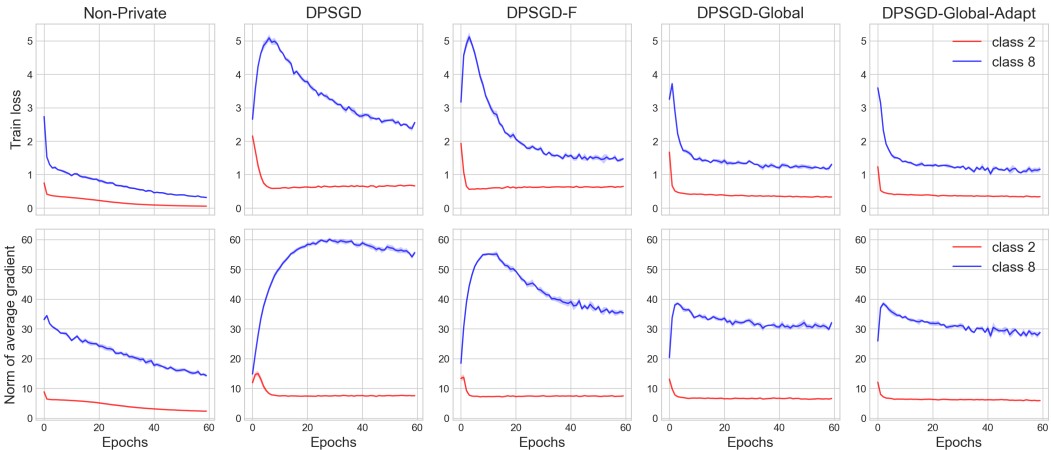

Figure 9: MNIST dataset. **Top**: Train loss per epoch. **Bottom**: $\|g_B\|$ averaged over batches per epoch.

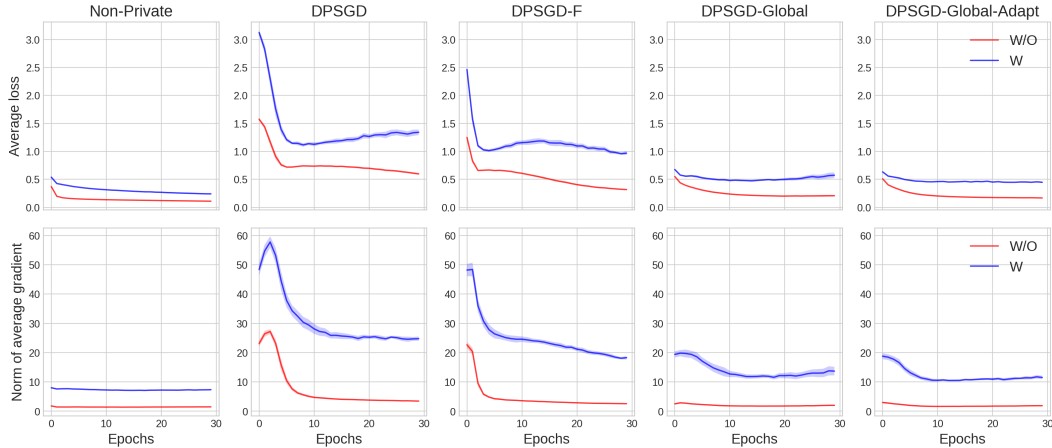

Figure 10: CelebA dataset. **Top**: Train loss per epoch. **Bottom**: $\|g_B\|$ averaged over batches per epoch.

To go along with the training curves shown for Dutch in Fig. 3, we present the same for Adult in Fig. 8, MNIST in Fig. 9, and CelebA in Fig. 10. The trends are consistent across datasets - whereas DPSGD produces large values and a large gap for the gradient norms and losses between protected groups, our method DPSGD-Global-Adapt reduces the values and gap at all stages of training.

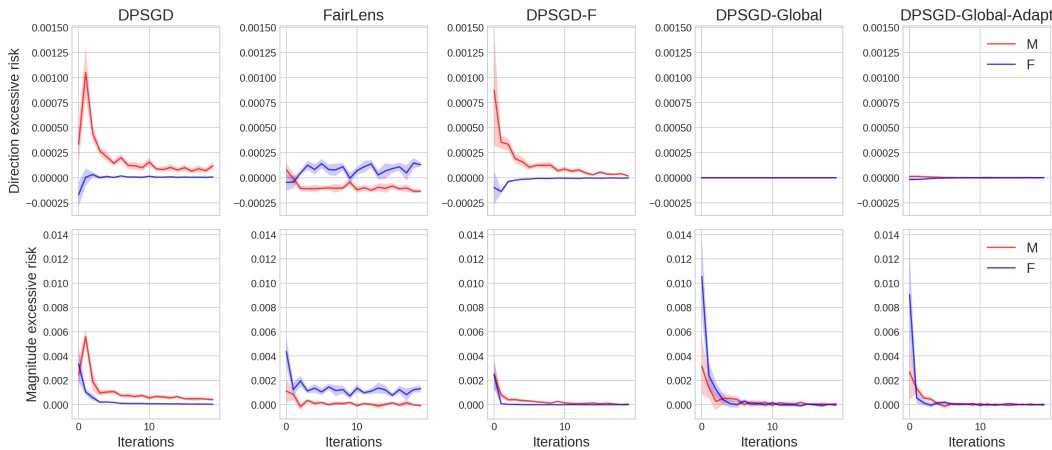

Figure 11: Dutch dataset. **Top**: Excessive risk due to gradient misalignment per group. **Bottom**: Excessive risk due to magnitude error per group.

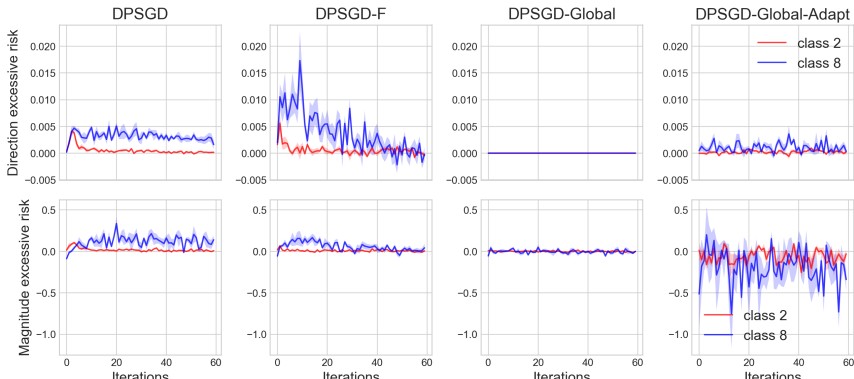

Figure 12: MNIST dataset. **Top**: Excessive risk due to gradient misalignment per group. **Bottom**: Excessive risk due to magnitude error per group.

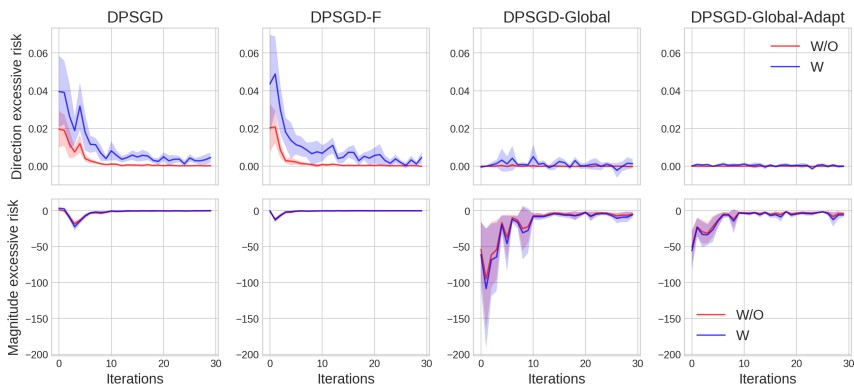

Figure 13: CelebA dataset. **Top**: Excessive risk due to gradient misalignment per group. **Bottom**: Excessive risk due to magnitude error per group.

We also present the values of terms $R_a^{\text{dir}}$ and $R_a^{\text{mag}}$ over training for Dutch in Fig. 11, for MNIST in Fig. 12, and CelebA in Fig. 13 as was done for Adult in Fig. 4. Both Global methods dramatically reduce $R_a^{\text{dir}}$ compared to DPSGD at the cost of larger $R_a^{\text{mag}}$. Comparing to the final training results where global methods also show the best performance, this provides further evidence for our hypothesis that gradient misalignment is the most significant cause of disparate impact in DPSGD.

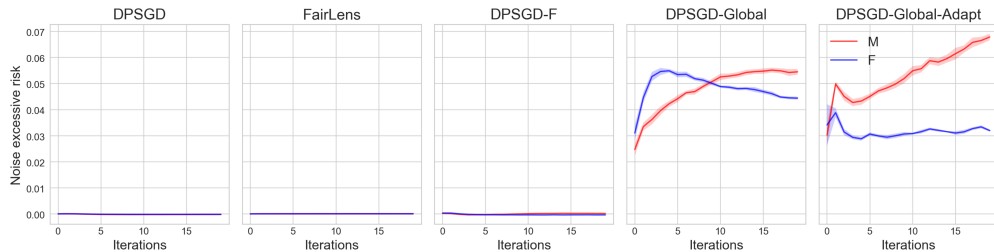

Figure 14: Excessive risk due to noise error per group for the Adult dataset

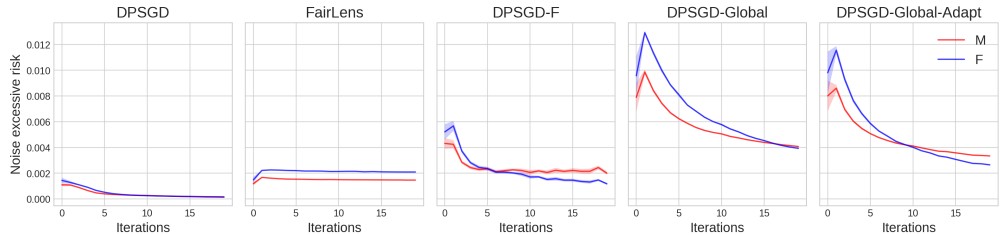

Figure 15: Excessive risk due to noise error per group for the Dutch dataset

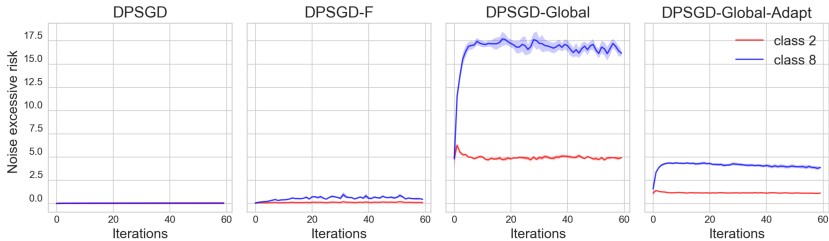

Figure 16: Excessive risk due to noise error per group for the MNIST dataset

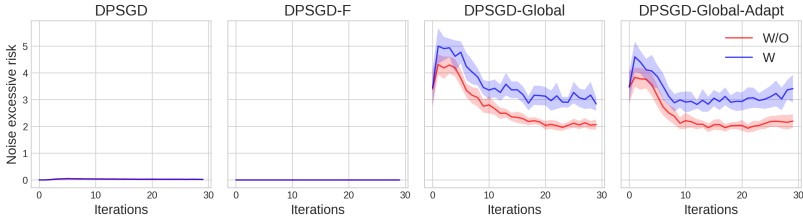

Figure 17: Excessive risk due to noise error per group for the CelebA dataset

We note that by tuning the learning rate in DPSGD-Global and DPSGD-Global-Adapt, there is a trade-off between magnitude error and noise error. Referring to Prop. 1, $R_a^{\text{noise}} = \frac{\eta_t^2}{2}\text{Tr}(H_\ell^a)C_0^2\sigma^2$, we see that the excessive risk due to noise is affected by the learning rate $\eta_t$, the noise multiplier $\sigma$, clipping bound $C_0$ and the trace of the Hessian for group $a$. In choosing a larger learning rate for the global methods to offset the magnitude error, we increase the noise error quadratically. Refer to the values of $R_k^{\text{noise}}$ over training for Adult in Fig. 14, Dutch in Fig. 15, MNIST in Fig. 16, and CelebA in Fig. 17. While the excessive risk due to noise is significantly larger for the global methods, these methods outperform all other private methods at the end of training, see Tables 2, 3, 4, 5. Gaussian noise adds zero bias and the errors it introduces tend to cancel out over the course of training. These observations further validate that direction error is the core cause of disparate impact, and minimizing gradient misalignment should be prioritized over other sources of unfairness.

