# OpenReview forum: "Disparate Impact in Differential Privacy from Gradient Misalignment"
_ICLR.cc/2023/Conference — ICLR 2023 notable top 25%_

### Official Review · Reviewer_Sern · 2022-10-20

**Confidence:** 4
**Correctness:** 3
**Technical Novelty And Significance:** 2
**Empirical Novelty And Significance:** 2
**Recommendation:** 6

**Clarity, Quality, Novelty And Reproducibility:**

The paper is well-written and the experimental results are strong. The problem and the basic building blocks are not new but they show by experiments that DPSGD-Global-Adapt improves DP-SGD Global and FairLens. Code is included for reproducibility.

**Strength And Weaknesses:**

Pro: The paper provides a new angle in understanding the disparate impact of DP-SGD - gradient misalignment. The mitigation solution, gradient scaling, is different from previous works like global clipping and [1]. The lower bound is much tighter in fig.5 than [1] - the two curves almost overlap, which is intriguing.

Cons: Despite the differences with previous work [1], the decomposition of excess risk into clipping bias and noising bias is not new. And that DP-SGD hurts fairness was already understood as the combination of per-example gradient clipping and gradient noise injection.

[1] Differentially Private Empirical Risk Minimization under the Fairness Lens

**Summary Of The Paper:**

This paper studies the fine-grained causes of unfairness in DPSGD and identifies gradient misalignment due to inequitable gradient clipping as the most significant source. They show that excess risk, the notion of fairness, can be decomposed into both non-private terms, clipping bias, noising bias according to previous work. Based on that work, they further analyze the gradient misalignment as a cause for unfairness. They show a lower bound for excessive risk gaps which is tight.

**Summary Of The Review:**

The paper provides an interesting view, gradient misalignment, for analyzing the unfairness of DP-SGD. The technique is not new and can not resolve the privacy-fairness tradeoff but empirically DP-SGD-Global-Adapt can work. The fairness problem in DP can be important and related to societal impact since many trustworthy aspects are connected and some applications might require both.

---

> ### Author Response · Authors · 2022-11-15
> **Initial Response**
>
>
> We thank you for your positive review! We are happy to address the points you brought up.
>
> 1. `Despite the differences with previous work [1], the decomposition of excess risk into clipping bias and noising bias is not new. And that DP-SGD hurts fairness was already understood as the combination of per-example gradient clipping and gradient noise injection.`
>
> We agree that [1] established the decomposition of excess risk into clipping bias and noising bias, and related work cited in Section 2 has clearly shown that clipping and noising cause unfairness. Our work goes further and isolates direction errors from clipping as the main cause of unfairness over and above noise and magnitude errors. By isolating this cause we were able to propose a way to mitigate unfairness that is more effective than previous work like [1].
>
> 2. `The technique is not new and can not resolve the privacy-fairness tradeoff but empirically DP-SGD-Global-Adapt can work.`
>
> We appreciate that you think our method is valuable, but we disagree that our method can not resolve the privacy-fairness tradeoff. Unlike the privacy-utility tradeoff which is fundamental, there is nothing to suggest that all private methods must be unfair. Indeed, Propositions 1 and 2 give a very clear understanding of exactly what causes unfairness in DPSGD - minimizing these causes will resolve the tradeoff. Our work is the first to suggest that the recently developed global clipping method should reduce unfairness, we provided theoretical analysis to demonstrate why, and tested these claims experimentally. In some of our experiments the final models were completely fair in that they had *zero* privacy cost gap and excessive risk gap (Tables 4, 5).
>
>
> [1] Tran, Dinh, Fioretto ``Differentially Private Empirical Risk Minimization under the Fairness Lens"

---

### Official Review · Reviewer_pvjf · 2022-10-23

**Confidence:** 4
**Correctness:** 3
**Technical Novelty And Significance:** 2
**Empirical Novelty And Significance:** 2
**Recommendation:** 6

**Clarity, Quality, Novelty And Reproducibility:**

This work does not suffer from clear clarity or reproducibility issues.

A few things are notable novel about this work:
1. Showing that clipping does not necessarily increase excessive risk (if gradients are aligned) is a nice and useful result.
2. Adapts existing private algorithm (DPSGD Global) for the purpose of reducing disparities


**Strength And Weaknesses:**

Strengths:
1. Nice, clean decomposition of magnitude vs direction error. It would be great to see this for other datasets as well.
2. Thorough comparison to previous techniques.

Weaknesses:
1. Authors compare only 1 epsilon per dataset.  It is unclear whether results also hold in other epsilon regimes.
2. Would like to see more discussion on why DPSGD and DPSGD-F diverge so much in Figure 3. Has proper hyperparameter tuning been done? It looks like DPSGD would further improve with more epochs?
3. It is unclear whether this analysis extends to more complex image datasets like CIFAR10 and CIFAR 100. It would be nice to see a discussion on why or why not.
4. How does noise compound issues of gradient misalignment? In Table 1, the authors compare the effects without noise. It is not immediately clear to me that similar results could hold for a fixed amount of noise.
5. The authors mention that DPSGD-Global has the potential to exacerbate disparate impact by discarding gradients, but I don't see this happening empirically in Figure 3 or Table 2. Please explain further.


**Summary Of The Paper:**

This paper examines the cause of disparate impact on unbalanced datasets (tabular and MNIST) and finds gradient misalignment to be the cause. Furthermore, they introduce techniques to mitigate these disparities.

**Summary Of The Review:**

There are several areas on which this paper can elaborate on. But overall, it is a useful contribution.

---

> ### Author Response · Authors · 2022-11-15
> **Initial Response**
>
>
> We thank you for your positive review and are glad you found our work to be a useful contribution. We aim to address your points below (paraphrased).
>
> 1. `Authors compare only 1 epsilon per dataset. It is unclear whether results also hold in other epsilon regimes.`
>
> While we presented tables of results for a single epsilon per dataset, our figures showing behaviour over training effectively give insight into other values of the privacy budget. By composition, each subsequent iteration of training gives a model with a larger epsilon. For example, in Figure 3 we can see the loss values for each algorithm at each epoch, from a very small epsilon after the first iteration to the final value (2.27 for this dataset). Each algorithm starts at elevated loss values, but global clipping methods achieve competitive losses for all epsilons, with smaller gaps indicating less disparate impact. If you think it would add value, we could overlay a plot of the epsilon value over training.
>
> 2. `Would like to see more discussion on why DPSGD and DPSGD-F diverge so much in Figure 3. Has proper hyperparameter tuning been done? It looks like DPSGD would further improve with more epochs?`
>
> It is entirely possible that DPSGD would improve with more iterations, but each iteration comes with a privacy cost. For this reason, the convergence speed of private algorithms is highly relevant, and the original Figure 3 showed that global clipping converges faster.
>
> DPSGD may be performing poorly because an MLP was not the best choice of model for the Dutch dataset. We have rerun the experiment using logistic regression models with each training algorithm, please see the updated Figure 3 and our general response for details on this.
>
>
> 3. `It is unclear whether this analysis extends to more complex image datasets like CIFAR10 and CIFAR 100.`
>
> Please see the general response for our experiments on a more complex image dataset, CelebA, which is significantly larger than either CIFAR 10 or 100.
>
> 4. `How does noise compound issues of gradient misalignment? In Table 1, the authors compare the effects without noise. It is not immediately clear to me that similar results could hold for a fixed amount of noise.`
>
> Noise generally has very little role to play in the unique properties of DPSGD. Per-sample clipping is by far the most important change compared to SGD. To see why, reconsider Figure 1 - adding Gaussian noise to the gradient increases the variance of SGD, but does not add bias. SGD already incorporates noise by selecting mini-batches, which similarly does not add bias, and SGD converges nicely. In contrast, per-sample gradient clipping adds bias. In particular, if there is a group in the data which is underrepresented or more complex, then that group tends to have larger gradient norms. Clipping will systematically reduce the influence of that group compared to other well-represented or simple groups. This type of bias can lead to suboptimal minima, and can prevent convergence - for example see Figure 3 where some private methods have elevated average gradient norms even at the end of training, showing that they are not settling down into a local minimum.
>
> 5. `The authors mention that DPSGD-Global has the potential to exacerbate disparate impact by discarding gradients, but I don't see this happening empirically in Figure 3 or Table 2. Please explain further. `
>
> As you know, in DPSGD-Global when a per-sample gradient has norm greater than $Z$, the gradient is set to zero. Hence, the algorithm has the potential to ignore groups in the data if $Z$ is not set appropriately. Obviously this is not beneficial for training, so we tried to set $Z$ appropriately in our experiments. We agree that it is not apparent in Figure 3 or Table 2 that DPSGD-Global is discarding too many gradients. In fact, Figure 4 shows this more clearly - DPSGD-Global has zero direction error which means no gradients are being discarded. Still, it has the *potential* to cause harm when $Z$ is not tuned. Our method improves this by adaptively setting $Z$, lessening the need for tuning.

---

### Official Review · Reviewer_ktiq · 2022-10-25

**Confidence:** 3
**Correctness:** 3
**Technical Novelty And Significance:** 2
**Empirical Novelty And Significance:** 3
**Recommendation:** 6

**Clarity, Quality, Novelty And Reproducibility:**

This paper is well-written. The statements in the paper are clear and concise. The novelty in the Propositions are not as significant as in the experiments. The code provided by the author is well-organized but I have not tried to reproduce the results.

**Strength And Weaknesses:**

Major strengths:
1. Proposition 3 is closely related to the focus of this paper on the disparate impact caused by gradient misalignment.
2. The authors have very detailed experiment results which supports their statements.

Minor strengths:
1. This paper is easy to follow and the algorithms are intuitive and can be implemented easily.

Major weaknesses:
1. The experiment results seem to be confusing. In Figure 3, the DPSGD is not working in terms of the train loss while the DPSGD-Global is even as good as the non-private result. Such results are not shown in Bu et al, 2021. In Figure 4, the DPSGD-Global and Adapt solves the direction excess risk, but the magnitude of this direction excess risk seems to be much smaller than the excess risk from the magnitude error. Additionaly, the excess risk from the noise error is analyzed in the supplementary material where I think the magnitude of them is much larger than the excess risk from the gradient misalignment and the magnitude error.
For example, for the Dutch dataset, I do not understand how the excess risk is break down into the three parts in the way that the authors claim the direction one contributes the most while has the smallest magnitude.
2. Proposition 3 does not show why $|R_a^{dir} - R_b^{dir}|$ is larger than $|R_a^{mag} - R_b^{mag}|$. It uses the inverse of the largest eigenvalue of the Hessian among different groups as an upper bound for the learning rate. It seems to be hard to satisfy such constraint. In Figure 5, this lower bound (I guess it is derived from the proof of Prop 3) is shown to be very good. Therefore, there is a lack of an explanation about the learning rate being used and the upper bound from the assumption of Prop 3.


**Summary Of The Paper:**

This paper identifies the gradient misalignment as the main reason why applying privacy-enhancing methods may lead to more unfairness. The authors use an existing method, DPSGD-Global, and a variant of it, DPSGD-Global-Adapt, to prevent misalignment and reduce this unfairness.

**Summary Of The Review:**

This paper shows detailed experimental results to support their statement that the unfairness is mainly caused by gradient misalignment. The interpretation of the results may have some problems which weaken the validity of the analysis.

---

> ### Author Response · Authors · 2022-11-15
> **Initial Response**
>
> We thank you for your objective review and are glad you found our method intuitive. We hope we can clarify some points based on your feedback below (paraphrased).
>
> 1. `In Figure 3, the DPSGD is not working in terms of the train loss while the DPSGD-Global is even as good as the non-private result. Such results are not shown in Bu et al, 2021.`
>
> We modelled our experimental suite on Xu et al. [1] for ease of comparison to prior work. In [1] the authors used logistic regression on Dutch, whereas in our original Fig. 3 an MLP network was used. We have repeated the Dutch experiment using logistic regression, and find similar results to [1] where DPSGD converges more consistently.
>
> In contrast Bu et al. [2] do not consider fairness, so they only plot loss curves for the entire dataset, not groups. Since the datasets differ, for example because we used an undersampled class 8 in MNIST, we do not expect our results to match.
>
> Please see our general response for details on the logistic regression experiment.
>
>
> 2. `In Figure 4, global clipping methods solve the direction excess risk, but increase magnitude excess risk, and noise excess risk. The magnitude and noise terms are even larger than the direction terms. Why do the authors claim direction errors contribute the most while having the smallest magnitude?`
>
> You make an astute observation - yes our results show that global clipping methods have large per-iteration excess risk from magnitude error and noise. On the other hand, their overall excess risk (measured at the end of training) is much lower! See Tables 2, 3, 4, and 5. We consider this as evidence for the main claim in our paper - direction errors matter more than other types of error for fairness over the course of training.
>
> To clarify, there are two ways of measuring excess risk, which we tried to emphasize at the end of Section 4. Proposition 2 and figures like Fig. 4 look at the difference in losses *over a single iteration*, comparing the optimal step using non-private SGD, to the step taken. Alternatively, Eq. 2 and Tables 2, 3, 4, and 5 look at the differences between models *over the course of training*, comparing a model fully trained with non-private SGD to a model fully trained with the private algorithms.
>
> As we say at the end of Section 4, ``the full impact of clipping errors may not be felt per-iteration,
> but only at convergence". Recall Figure 1 - magnitude errors only slow down convergence but do not change the trajectory of descent; noise errors introduce some variance, but zero bias, and hence they tend to cancel out over training; direction errors introduce real bias and tend to build up over training leading to suboptimal convergence. We have added to the discussion around Figure 1 to make this intuition more evident. Our experimental results using both methods of measuring excess risk support this high level picture and show that direction error matters most to overall fairness, even though it has the smallest magnitude per-iteration.
>
> 3. `Proposition 3 does not show why` $|R_a^{\text{dir}}-R_b^{\text{dir}}|$ `is larger than` $|R_a^{\text{mag}}-R_b^{\text{mag}}|$.
>
> We agree that Proposition 3 does not show that $|R_a^{\text{dir}}-R_b^{\text{dir}}|$ is larger than $|R_a^{\text{mag}}-R_b^{\text{mag}}|$, but this was not the purpose of the result. The premise of our work is that $R_a^{\text{dir}}$ matters most for fairness over the course of training, and Proposition 3 gives a simpler condition to diagnose when disparate impact from direction errors is occurring.
>
> 4. `Proposition 3 uses the inverse of the largest eigenvalue of the Hessian among different groups as an upper bound for the learning rate. There is a lack of an explanation about the learning rate being used in Figure 5 and the upper bound from the assumption of Prop 3.`
>
> Figure 5 shows the lower bound on $|R_0^{\text{dir}}-R_1^{\text{dir}}|$, which is just the difference between sides of the inequality in Prop 3. When the lower bound is positive, $|R_0^{\text{dir}}-R_1^{\text{dir}}|$ must also be positive indicating disparate impact. As you rightly point out, the bound requires the learning rate to be less than the inverse of the maximum eigenvalue of the Hessian calculated for either group. We have verified that this requirement is met during training for the MNIST dataset. If you believe it would add to the paper, for the final version we can overlay a plot of the maximum Hessian eigenvalue on Figure 5. Also, experimental details such as learning rates are carefully laid out in App. B.2.
>
> [1] Xu, Du, Wu ``Removing Disparate Impact on Model Accuracy in Differentially Private Stochastic Gradient Descent"
>
> [2] Bu, Long, Su. ``On the Convergence of Deep Learning with Differential Privacy."

---

> > ### Comment · Reviewer_ktiq · 2022-11-17
> > **The impact from the direction errors and magnitude errors to the overall excess risk is still a conjecture.**
> >
> > 1. Thanks for the added experiment for the Dutch dataset. It looks good to me.
> >
> > 2. Got it. So, the main claim is about the overall excess risk at the end of the training, and Propositions 1, 2, and 3 and Figure 4 are for each step's overall excess risk. You said that 'the full impact of clipping errors may not be felt per iteration, but only at convergence'; therefore, the only evidence to support your main claim is Tables 2, 3, 4, and 5. I understand that you choose to show the theoretical and experimental results for per-iteration performance to find the fundamental reason/evidence why the directional excess risk contributes more than the magnitude excess risk, but the logic chain is still missing a critical part: how the impact of clipping comes from per iteration gives the final impact at convergence. I see you have added a sentence at the end of Section 6.2, 'Direction errors introduce bias which accumulates, whereas magnitude errors do not alter the convergence path, and noise errors add zero bias and tend to cancel out.' However, I guess this is only a conjecture, not a verified result, right?
> >
> > 3. Got it. Then my main question is in point (2) above.
> >
> > 4. Thanks for the explanation. As you have mentioned, 'We have verified that this requirement is met during training for the MNIST dataset.', please consider adding this verification in the appendix (maybe putting this result in the main text is too much, but it is nice to know that the experiments actually satisfy the assumptions in the propositions.)

---

> > > ### Author Response · Authors · 2022-11-18
> > > **We provide the reader theoretical results, experimental evidence, and intuition**
> > >
> > > It sounds like most of your concerns have been clarified and the main remaining point to clear up is below.
> > >
> > > 2. `... You said that 'the full impact of clipping errors may not be felt per iteration, but only at convergence'; therefore, the only evidence to support your main claim is Tables 2, 3, 4, and 5. ... the logic chain is still missing a critical part: how the impact of clipping comes from per iteration gives the final impact at convergence. ... I guess this is only a conjecture, not a verified result, right?`
> > >
> > > Your summary of the work is very good. We do not yet have a theoretical understanding that connects per-iteration excess risk to the overall excess risk at the end of training. In other words, we do not have a theoretical understanding of how excess risk accumulates iteration-by-iteration to reach some final value at the end of training. We are not aware of any work that can quantify, in a simple way, the overall excess risk of using one algorithm over another - in this case DPSGD over SGD. Such a result for DPSGD would be extremely interesting, but is not within the scope of our work.
> > >
> > > Our evidence to support that gradient misalignment is the most significant cause of disparate impact overall does come from Tables 2, 3, 4, and 5. The global clipping methods that quantitatively have less per-iteration excess risk from gradient misalignment (e.g Fig. 4) also have less overall excess risk (i.e. disparate impact).
> > >
> > > To emphasize the intuition introduced at the start of Section 4, we added the statement in Section 6.2: `Direction errors introduce bias which accumulates, whereas magnitude errors do not alter the convergence path, and noise errors add zero bias and tend to cancel out'. This is again meant to provide intuition about the training dynamics, and is not rigorously formulated.
> > >
> > > However, it is a fact that the noise added in DPSGD and variants is zero mean and sampled i.i.d. for each iteration, so it adds zero bias. By definition, magnitude errors involve only a change in scale of the gradient, so the descent update still acts in the correct direction. Note that the clipped gradients will *usually* be smaller than optimal, not larger, so this should only slow convergence and not cause divergence. Finally, in the fairness context if there is a group in the data which is underrepresented or more complex, then that group *usually* has larger gradient norms. Larger norms mean that group will **systematically** be clipped causing direction errors, which is the bias we mentioned. You probably notice the emphasis on "usually". We know that these intuitions do not hold up in every case, so we have not attempted to provide guarantees. Nevertheless, we think that the intuition provided to the reader is helpful.
> > >
> > >
> > > 4. `Thanks for the explanation. As you have mentioned, 'We have verified that this requirement is met during training for the MNIST dataset.', please consider adding this verification in the appendix`
> > >
> > > Absolutely, for the camera-ready version of the paper we will add this verification.

---

> > > > ### Comment · Reviewer_ktiq · 2022-11-18
> > > > **Thanks for the prompt explanation!**
> > > >
> > > > `However, it is a fact that the noise added in DPSGD and variants is zero mean and sampled i.i.d. for each iteration, so it adds zero bias.`
> > > > It adds zero bias for each iteration, but there still could be bias over time since, in general, $\mathbb{E}[f(\tilde{X})] \neq f(\mathbb{E}[\tilde{X}])$, therefore, if we think $f$ as 'parameter update and gradient operation' and $X$ as 'the gradient value', even we know $\mathbb{E}[\tilde{X}] = X$, we do not know whether there is a bias in the end.
> > > >
> > > > `By definition, magnitude errors involve only a change in scale of the gradient, so the descent update still acts in the correct direction.`
> > > > Similar to the argument above, we do not know how the next step gradient changes when the magnitude of the previous gradient-update changes.
> > > >
> > > > I understand the difficulty here and really appreciate your straightforward explanation. I would raise my recommendation score to 6.

---

### Official Review · Reviewer_tDQn · 2022-10-25

**Confidence:** 4
**Correctness:** 4
**Technical Novelty And Significance:** 3
**Empirical Novelty And Significance:** 3
**Recommendation:** 8

**Clarity, Quality, Novelty And Reproducibility:**

The writing in this paper is quite clear and easy to follow. I have a minor suggestion above, but overall, each section is written and presented clearly. This work builds on insights from previous analyses of DP-SGD; however, it brings new information to light. Overall, the results show that gradient alignment is a key cause of disparate impact.

**Strength And Weaknesses:**

### Strengths

- This paper refines the cause of disparate impact in SGD, which is an important problem.

- The experiment and analysis presented in Section 4 clearly show the main take home of this paper.

- The proposed modification to DP-SGD is quite simple and straightforward. The proposed optimizer can be easily plugged-in to other settings.

### Weaknesses
- One of the benefits of the proposed DP-SGD-Adapt global is that it does not need group identifiers. However, it is unclear whether the methods that use group identifiers lead to models with better utility. While the comparison is not 1:1 here, it would still be good to know how these results compare to when one actually takes into account group identifiers.

- The datasets considered the paper are small and somewhat toyish even though these are the datasets that were used in previous papers. It is unclear how this proposal scales to larger models and bigger setups. However, I think future work can likely address these issues.

- The theorems and analysis presented applies or explicitly assume convexity or in some cases that the loss is twice differentiable. It is nice that the analysis generalizes to model classes that do not satisfy these assumptions but this issue could be better highlighted.

- This work highlights gradient misalignment as a key cause of disparate impact, but it is hard to judge how much that is more important than say group difficulty as measured by the trace of the hessian of the loss per group. It could be that more difficult to learn groups have hessian-trace values that are substantially different than other groups.

-  (Minor) The term 'excessive risk' is used quite a bit in a number of places in the paper. I believe it should be 'excess risk'.

**Summary Of The Paper:**

This paper conducts a fine-grained analysis of DP-SGD to show that gradient misalignment is  a principal cause of the disparate impact that occurs when DP-SGD is applied to a dataset. This analysis compares two key components of the DP-SGD: clipping and noise addition. Ultimately, this paper shows that clipping further impacts and exacerbates gradient misalignment, which greatly increases the disparate impact of DP-SGD. This insight is nicely illustrated in a simple experiment that keeps the noise level in DP-SGD fixed, but changes the level of gradient alignment to show that when the gradient is more misaligned, it leads to more disparate results. The fix proposed by this paper is to  scale down all per-sample gradients in a batch by the same amount. The improved results are demonstrated experimentally on the MNIST and adult datasets.

**Summary Of The Review:**

This paper provides new insights on an important problem. The paper also demonstrates empirically improvements based on a modified DP-SGD algorithm that reduces disparate impact.

---

> ### Author Response · Authors · 2022-11-15
> **Initial Response**
>
> We thank you for your thorough review and thoughtful feedback, and are glad you appreciated the new information our work brings to light. Please see our points below, which we hope address your concerns (paraphrased):
>
> 1. `One of the benefits of the proposed DPSGD-Global-Adapt is that it does not need group identifiers. However, it is unclear whether the methods that use group identifiers lead to models with better utility.`
>
> We compared our proposed method to two existing solutions that do use group identifiers [1, 2] to provide better fairness. Our experiments showed that our method had a statistically significant improvement in accuracy over [1] and [2] for each dataset tested, on top of being more fair.
>
> As you point out, it is a major benefit that our approach does not require group identifiers, because collecting that data may further infringe on an individual's privacy, or may be outright impossible due to regulatory concerns.
>
> 2. `The datasets considered in the paper are small. It is unclear how this proposal scales to larger models and bigger setups.`
>
> As you noticed, we presented results on the same datasets as past work [1] in this area for ease of comparison, but they were all small datasets. See our general response for new results on a larger dataset.
>
> 3. `The analysis presented assumes convexity or that the loss is twice differentiable, this issue could be better highlighted.`
>
> Yes, our mathematical results in Prop. 2 and 3 assume that the loss function is twice differentiable, and Prop. 3 also requires convexity. Differentiability of the loss function is a mild requirement, since most loss functions in machine learning are designed to be differentiable for backpropagation. Convexity of the loss function with respect to model parameters does hold for common approaches like linear or logistic regression, but not usually for neural networks. We have added discussion about these limitations in the appendix alongside the proofs.
>
> 4. `It is hard to judge how much more important gradient misalignment is than group difficulty as measured by the trace of the hessian of the loss per group.`
>
> We agree that this is a difficult question worthy of further research, and was touched upon in [2]. In DPSGD, the trace of the hessian plays a role in unfairness by controlling excessive risk from noise addition (see Prop. 1 in our work which was derived in [2]). While clipping, not noise, was the main focus of our work, we did compare the size of the trace hessian term in App. B.6, Figures 13, 14, 15, and 16 in the updated version. We showed that global clipping methods have increased trace Hessian terms per-iteration, but nevertheless cause less overall disparate impact (Tables 2, 3, 4, 5). We take this as evidence that gradient misalignment is more important than group difficulty, since the global clipping method specifically reduces gradient misalignment and performs better.
>
> 5. `The term 'excessive risk' should be 'excess risk'.`
>
> Thank you for pointing out this potential error. We have seen both terms used (e.g. [2]), but perhaps `excess risk' is more standard.
>
> [1] Xu, Du, Wu ``Removing Disparate Impact on Model Accuracy in Differentially Private Stochastic Gradient Descent"
>
> [2] Tran, Dinh, Fioretto ``Differentially Private Empirical Risk Minimization under the Fairness Lens"

---

> > ### Comment · Reviewer_tDQn · 2022-11-17
> > **Thanks for the response**
> >
> > I have read the response of my original review as well as the others. The clarifications are quite informative.
> >
> > - It is quite surprising to me that the new optimizer results in models with better performance than those that take into account group information. Actually, that is somewhat shocking to me since there are some results in the literature that suggests that a classifier that takes into account group information should not underperform one that doesn't use that information. Which method in Table 1 or 2 is the one that takes into account group information?
> >
> > - I am confused about the comment on trace. Shouldn't it be that if the trace Hessian increases then indicates that the group is more difficult to learn. In that case, increase in trace Hessian terms per-iteration during training would explain why there is disparate impact in these methods. Also, we line in the plots you mentioned actually correspond to the estimate of the Hessian trace? The y axis for those plots say excess risk due to noise. Was the trace estimated for these groups or are you using the noise as estimate of the Hessian trace.
> >
> > Overall, non of these are disqualifying. I am trying to understand the reasoning behind these statements.

---

> > > ### Author Response · Authors · 2022-11-18
> > > **Group labels and Hessian traces**
> > >
> > > We're glad that this discussion has been informative and can pursue further clarifications.
> > >
> > > 1. `Which method in Table 1 or 2 is the one that takes into account group information?`
> > >
> > > In Tables 2 and 3 (updated version) the method DPSGD-F [1] uses group label information to set the clipping threshold C differently for each group.
> > >
> > > In addition, Tables 4 and 5 also display ``FairLens" [2], which adds regularization terms to the loss function. The regularization terms involve quantities computed on subsets of the batch corresponding to each group.
> > >
> > > For your convenience, you can find our review of the DPSGD-F and FairLens methods in App B.5.
> > >
> > > We are interested in learning more about the results you mention and would greatly appreciate if you could share references with us.
> > >
> > >
> > > 2. `I am confused about the comment on trace. Shouldn't it be that if the trace Hessian increases then indicates that the group is more difficult to learn. In that case, increase in trace Hessian terms per-iteration during training would explain why there is disparate impact in these methods. Also, we line in the plots you mentioned actually correspond to the estimate of the Hessian trace? The y axis for those plots say excess risk due to noise. Was the trace estimated for these groups or are you using the noise as estimate of the Hessian trace.`
> > >
> > > The plots we mentioned show the quantity $R_a^{noise} = \frac{\eta}{2}\text{Tr}(H^a)C_0^2 \sigma^2$ from Prop. 1, which is the excess risk due to the addition of noise for DP. In these plots, for a given method (e.g. DPSGD-Global) the factors $\eta$, $C_0$, and $\sigma$ are constants, so the shape of the curves over iterations comes from the $\text{Tr}(H^a)$ factor. $\text{Tr}(H^a)$ is computed from data per group; it is not estimated from the noise/excess risk. The method of computation is through the Hutchinson trace estimator and Hessian-vector products (see App B.3 for implementation details).
> > >
> > > An increased Hessian trace, related to the group difficulty, certainly explains one source of disparate impact in these methods. But it is not the only source. Our work isolates the trace Hessian term ($R_a^{noise}$) from the two other sources $R_a^{dir}$ and $R_a^{mag}$, and our experiments show that $R_a^{dir}$ matters most. The global clipping methods specifically minimize $R_a^{dir}$, and they consistently have the least disparate impact.
> > >
> > > [1] Xu, Du, Wu ``Removing Disparate Impact on Model Accuracy in Differentially Private Stochastic Gradient Descent"
> > >
> > > [2] Tran, Dinh, Fioretto ``Differentially Private Empirical Risk Minimization under the Fairness Lens"

---

> > > > ### Comment · Reviewer_tDQn · 2022-11-18
> > > > **Interesting**
> > > >
> > > > Hello,
> > > >
> > > > 1. Group information: I now see that I mistook what it means to 'use' group information. I assumed that by use, these formulations decouple the classifiers for different groups and then train those group-based classifiers separately. But that is not the case here. I would say it is still somewhat surprising and nice that the new optimizer outperforms the group-based clipping variant. I guess this improvement speaks to the point about the important of gradient alignment versus clipping, which is the main point in this paper. One paper that has theoretical result I meant is "To Split or not to Split: The Impact of Disparate Treatment in Classification", which lower bounds the performance of a decoupled classifier versus a group-based one. However, this is all orthogonal to the discussion here since at the end the final model learnt is still a single classifier.
> > > >
> > > >
> > > > 2. Got it on the hessian trace now. The issue about group difficulty is not the main point of this paper, so I don't want to belabor this. To understand where these different trace terms come from, I assume Theorem 2 and it details in the Tran et. al. paper is what I should look into in more detail.
> > > >
> > > > Thanks

---

### Author Response · Authors · 2022-11-15
**General Response to All Reviewers**

We thank all four reviewers for their insightful feedback and constructive reviews. We are pleased to see that reviewers thought our work tackled an important problem (**tDQn**, **Sern**), that our experiments were thorough and clear (**tDQn**, **ktiq**, **pvjf**, **Sern**), and that the paper was well-written (**tDQn**, **ktiq**, **pvjf**, **Sern**).

We have updated our manuscript and supplementary material to address the points raised by reviewers. References in these responses now point to the new version. Since several comments were similar across reviewers, we address those points here. Detailed responses are also provided to each reviewer below.

**Larger datasets**: Reviewers **tDQn** and **pvjf** expressed a desire to see how our proposal scales to larger and more complex datasets, as the initial submission only contained experiments on MNIST and tabular datasets. We have repeated our experimental methodology for the much larger CelebA dataset with the classification problem of predicting gender. The ``protected group" is defined as those images containing eyeglasses, which is a minority group and is empirically harder to classify correctly. Our results in the new Table 3 once again show that among all private methods DPSGD-Global-Adapt has the best accuracy, privacy cost gap, loss, and excessive risk gap. Training losses and average gradient norms for CelebA are shown in Figure 9, while the three types of excessive risks are in Figures 12 and 16.


**DPSGD's convergence**: Reviewers **ktiq** and **pvjf** asked about the poor convergence of DPSGD in the original Figure 3. Compared to Xu et al. [1] in which the authors used logistic regression on Dutch, in our original Figure 3 an MLP network was used. The reviewers may have observed that the training loss increased for part of the DPSGD training, especially for the underrepresented ``F" group. We see this as an instability of DPSGD and a source of unfairness caused by its imbalanced clipping.

To ensure that the outcomes we claimed are a result of algorithmic improvements, and not because of problems in training MLP networks, we repeated the Dutch experiment using logistic regression and have updated those results in the new submission. In the new Figure 3 all methods have smoothly decreasing training loss, but global clipping methods are the most similar to non-private training, meaning the cost of adding privacy is small at all points during training. We also see that DPSGD's average gradient norm now decreases, but remains elevated and with a larger gap compared to global clipping. The remainder of results on Dutch are in the updated Table 5, Figure 10, and Figure 14. Our conclusions in the paper remain unchanged in light of this update.

We have updated our code in the supplemental material with these new experiments for reproducibility.

[1] Xu, Du, Wu ``Removing Disparate Impact on Model Accuracy in Differentially Private Stochastic Gradient Descent"

---

### Author Response · Authors · 2022-11-17
**Reminder for discussion**

We kindly remind our reviewers that the discussion period comes to an end in one day. Do you have any further concerns we can address?

---

### Decision · Program_Chairs · 2023-01-20

**Decision:**

Accept: notable-top-25%

**Justification For Why Not Higher Score:**

While the idea is quite interesting, there are not many deep insights to be taken away from this.

**Justification For Why Not Lower Score:**

The idea is interesting enough in its own right to justify airing to a larger audience.


**Metareview: Summary, Strengths And Weaknesses:**

The paper has an interesting idea, and all the reviews agree that it's of good quality.


**Note From Pc:**

if the above contains the word "oral" or "spotlight" please see: "oral" presentation means -> notable-top-5% and "spotlight" means -> notable-top-25%. As stated in our emails, we are disassociating presentation type from AC recommendations